# Timely regulated sorting from early to late endosomes is required to maintain cerebellar long-term depression

Taegon Kim[1], Yukio Yamamoto[1] & Keiko Tanaka-Yamamoto[1]

An important feature of long-term synaptic plasticity is the prolonged maintenance of plastic changes in synaptic transmission. The trafficking of AMPA-type glutamate receptors (AMPARs) is involved in the expression of many forms of synaptic plasticity, yet the subsequent events accomplishing the maintenance of plastic changes in synaptic AMPAR numbers are not fully understood. Here, we find that maintenance of cerebellar long-term depression results from a reduction in the number of AMPARs residing within endocytic recycling pathways. We then develop a genetically encoded, photosensitive inhibitor of late endosome sorting and use this to discover that initial maintenance of long-term depression relies on timely regulated late endosome sorting, which exhibits a threshold as well as switch-like behavior. Thus, our results indicate that recycling AMPAR numbers are reduced by a switching machinery of transient late endosome sorting, and that this process enables the transition from basal synaptic transmission to long-term depression maintenance.

[1] Center for Functional Connectomics, Korea Institute of Science and Technology (KIST), Seoul 136-791, Republic of Korea. Correspondence and requests for materials should be addressed to Y.Y. (email: yukio.kist@gmail.com) or to K.T.-Y. (email: keikoyamat@gmail.com)

Brain circuit function relies on the strength of synaptic transmission. In order for excitatory synaptic transmission to be stable, the number of postsynaptic α-amino-3-hydroxy-5-methyl-4-isoxazolepropionic acid (AMPA)-type glutamate receptors (AMPARs) must be constant despite undergoing continuous trafficking[1]. However, during long-term synaptic plasticity, synaptic transmission is dynamically regulated by changes in postsynaptic AMPAR levels[2]. It is not yet known how AMPAR trafficking switches between basal, constitutive levels, and the altered levels underlying long-term synaptic plasticity.

In the cerebellum, simultaneous activity at parallel fiber (PF) and climbing fiber synapses into Purkinje cells (PCs) induces long-term depression (LTD) at the synapses between PFs and PCs[3]. The simultaneous synaptic activity triggers transient increases in postsynaptic calcium concentration[4], with the increased calcium activating a positive feedback loop that includes protein kinase C (PKC) and mitogen-activated protein kinase (MAPK)[5–7]. This loop causes sustained activation of PKC, which in turn phosphorylates AMPARs and yields net internalization of postsynaptic AMPARs via clathrin-mediated endocytosis[8–10]. However, even though LTD was interrupted by a PKC inhibitor applied soon after LTD induction, LTD was not affected by the inhibitor applied after LTD is triggered, suggesting that enhanced internalization of AMPARs by active PKC is no longer required after LTD is triggered[5]. In addition, application of a PKC inhibitor 20 min after LTD induction can result in two distinct responses: either sustained depression, leading to LTD, or transient depression, which allows synaptic transmission to recover to basal levels[11]. These results indicate that LTD consists of at least two components distinguished by the responses to the timed application of PKC inhibitor, a reversible expression phase and an irreversible maintenance phase. Thus, there seems to be a switch, downstream from the PKC-MAPK positive feedback loop, that determines whether synaptic transmission is maintained in the basal state or makes a transition to a sustained depressed state that underlies LTD.

Many studies have shown that AMPAR trafficking plays a crucial role in both long-term potentiation (LTP) and LTD, including cerebellar LTD[8]. As part of their trafficking, AMPARs can be internalized; afterwards, AMPARs can be recycled back to the synaptic membrane, through a pathway that involves early endosomes (EE) and recycling endosomes, or they can be sorted from EE to late endosomes (LE) and lysosomes[1]. Sorting from EE to LE is required for LTD in hippocampal pyramidal neurons, thereby reducing the number of AMPARs[12, 13]. Considering that LE sorting is downstream of AMPAR internalization, this sorting might be part of the switch mediating the transition from the basal level to the depressed level during maintenance of LTD. However, the properties and timing of LE sorting during LTD are completely unknown.

In the present study, we develop a photosensitive inhibitor of LE sorting by connecting a light-oxygen-voltage (LOV) domain[14] to a dominant-negative mutant (T22N) of Rab7 (Rab7TN). We use this to investigate how and when LE sorting occurs during cerebellar LTD. Our results demonstrate that LTD relies on transient LE sorting that occurs around the time of synaptic depression reaching a maximum. In addition, our experimental results, together with computational models, suggest that this transient sorting serves as a switch to reduce the number of AMPARs in the synaptic recycling loop, so that the depressed level of AMPARs can be maintained without altering the rate of AMPAR trafficking. Thus, our study demonstrates the properties and timing of the switch underlying maintenance of cerebellar LTD and presumably of other forms of LTD arising from the reduced number of synaptic AMPARs.

## Results

**Reduced number of recycling AMPARs during LTD maintenance.** We first addressed a question whether AMPAR trafficking is altered during LTD maintenance, by introducing tetanus toxin (TeTx, 200 nM) through a patch pipette while recording excitatory postsynaptic currents (EPSCs) evoked by electrical stimulation of PF axons (PF-EPSCs) in PCs at basal state or during LTD maintenance (Fig. 1a). TeTx causes a reduction of PF-EPSCs[15], which results from the reduction in synaptic AMPAR numbers by TeTx interrupting AMPAR reinsertion into the plasma membrane. Therefore, the time constant of the reduction in PF-EPSC amplitude represents the speed of constitutive AMPAR internalization, while the degree of reduction is proportional to the number of AMPARs within the recycling loop. The LTD stimulation was applied by treating slices with 50 mM K⁺ and 10 μM

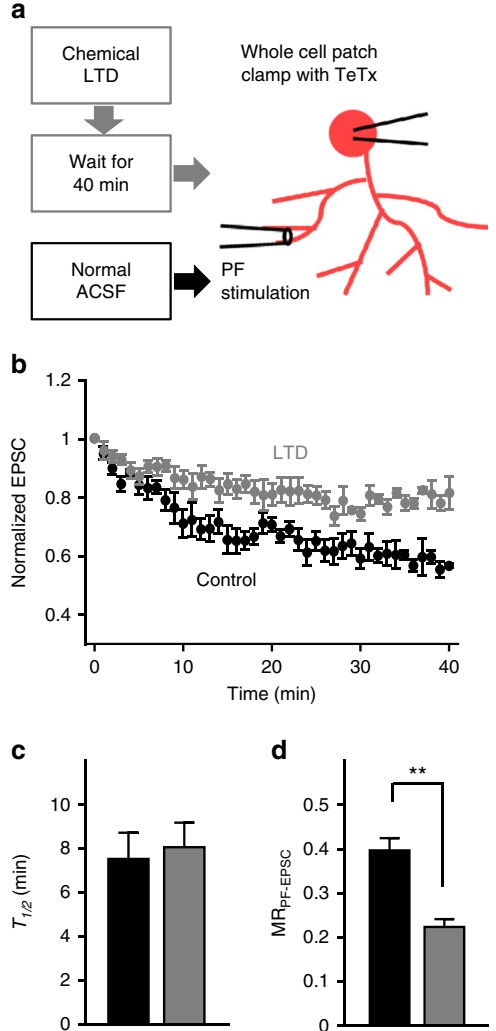

**Fig. 1** Reduction in the number of recycling AMPARs during LTD maintenance. **a** Design of the experiments to investigate AMPAR recycling using TeTx. **b** Time course of changes in PF-EPSC amplitudes by the application of TeTx through patch pipette. The time right before the start of PF-EPSC reduction was defined as 0 min. Cerebellar slices treated with K-glu (*gray circles*, LTD, $n = 5$) or kept in normal ACSF (*black circles*, control, $n = 6$) were used. PF-EPSC amplitudes are normalized to their level during the first 1 min. **c, d** $T_{1/2}$ **c** and MR$_{\text{PF-EPSC}}$ **d** in TeTx-dependent PF-EPSC reduction. **P < 0.01, Student's *t*-test. Exact *P* values for the data sets in this and subsequent figures are provided in Supplementary Table 2

glutamate (K-glu) for 5 min, which was indeed shown to induce LTD similar to that evoked by synaptic activity in terms of amount (~30%) and time course[5, 16]. In addition, the K-glu-induced LTD appears to be based on the same signaling mechanisms employed on LTD evoked by synaptic activity, because the K-glu treatment occluded subsequent induction of LTD by synaptic activity[17]. Recordings were then made around 40 min after K-glu treatment, to test PF-EPSCs during LTD maintenance. As expected, introducing TeTx into PCs caused a gradual reduction in PF-EPSCs, both in control PCs as well as those in which LTD had been induced (Fig. 1b, Supplementary Fig. 1a). The time to reach half-maximum reduction in PF-EPSCs ($T_{1/2}$), extracted by fitting an exponential decay function, were not significantly different in control and LTD conditions (Fig. 1c). On the other hand, the maximum reduction in PF-EPSCs ($MR_{PF-EPSC}$) during LTD maintenance was significantly less than in control conditions (Fig. 1d). These results indicate that while the rate of AMPAR internalization is similar in baseline and LTD maintenance, the number of AMPARs in the recycling loop is reduced during LTD maintenance.

**A quantitative model for AMPAR trafficking**. To confirm our interpretation, we built a simplified model of AMPAR trafficking, which consisted of four AMPAR pools: mobile, stable synaptic pools, extrasynaptic, and intracellular pools. (Fig. 2a). AMPARs undergo linear transitions between mobile synaptic, extrasynaptic, and intracellular pools, based on observations that AMPARs are not directly inserted into or internalized from synaptic area[18, 19]. We first considered two possible mechanisms for LTD maintenance, either (1) reducing total AMPAR numbers in the recycling loop (reduced $N_{tot}$ in Fig. 2a) or (2) enhancing the transition from extrasynaptic to intracellular pools (increased $k_{endo}$ in Fig. 2a). Changes in either parameter could indeed reduce EPSC amplitude that is proportional to the number of AMPARs in mobile and stable synaptic pools (Fig. 2b, top panels). TeTx treatment was then mimicked by preventing the transition from the intracellular to extrasynaptic compartments (reduction in $k_{exo}$ in Fig. 2a). This could reproduce the TeTx-dependent reduction of baseline PF-EPSCs (Fig. 2c, black line). To model the effects of $k_{exo}$ inhibition during LTD maintenance, we measured $T_{1/2}$ (Fig. 2b, middle panels) and $MR_{PF-EPSC}$ (Fig. 2b, bottom panels) of $k_{exo}$ inhibition-mediated EPSC reduction under conditions of reduced $N_{tot}$ or increased $k_{endo}$. The reduction of $N_{tot}$ led to changes in $MR_{PF-EPSC}$, but not in $T_{1/2}$, whereas an increase in $k_{endo}$ caused changes in both $MR_{PF-EPSC}$ and $T_{1/2}$ (Fig. 2b), indicating that the reduction of $N_{tot}$ could reproduce experimentally obtained results. We further confirmed that the model with a specific value of reduced $N_{tot}$ (a in Fig. 2b) resulted in 35% EPSC reduction (top panel in Fig. 2b), equivalent to K-glu-induced LTD[5], and fitted well to the experimental results of TeTx-induced EPSC reduction during LTD maintenance (Fig. 2c–e). In contrast, increasing $k_{endo}$ (b in Fig. 2b) to reproduce the decline in EPSC amplitude during LTD, changed $T_{1/2}$ during TeTx treatment (Fig. 2c–e), unlike the experimental observations. Thus, the model confirmed that LTD maintenance is associated with a reduction in the number of AMPARs within the recycling loop, rather than an increase in the rate of AMPAR internalization.

For the abovementioned analyses, we assumed that $T_{1/2}$ of PF-EPSC reduction by TeTx solely relies on the rate of AMPAR internalization, but not gradual blockade of exocytosis due to the diffusion of TeTx, even though the internal solution was gradually diffused into PCs (Supplementary Fig. 1a). This assumption is based on two series of studies. First, higher concentration of TeTx did not seem to accelerate the speed of

TeTx-mediated EPSC reduction in PCs[15], suggesting that TeTx blocks exocytosis nearly in an all-or-none manner. Second, antibody-feeding experiments in cultured pyramidal neurons estimated the rate of AMPAR internalization at the basal state as $T_{1/2}$ of about 7 min[1, 20], which is similar to $T_{1/2}$ of PF-EPSC reduction (7.5 min) in the current study. However, the diffusion of TeTx may slightly slow the blockade of exocytosis, and $T_{1/2}$ of PF-EPSC reduction may be partially affected by the gradual blockade. To test whether our interpretation can be made even in case of gradual blockade of exocytosis, we included the gradual blockade with a half decay period of 5 min in the model (Supplementary Fig. 1b). The TeTx-induced EPSC reduction at the basal state could be reproduced by this model with a modification of $k_{endo}$ from 0.13 to 0.55 min$^{-1}$ (Supplementary Fig. 1c), resulting in the estimation of the basal AMPAR endocytosis rate as $T_{1/2}$ of 2.8 min, 2.5 times faster than reported rate[1, 20]. Nevertheless, this model still supports our interpretation: the reduction of $N_{tot}$ led to changes in $MR_{PF-EPSC}$, but not in $T_{1/2}$, whereas an increase in $k_{endo}$ caused changes in both $MR_{PF-EPSC}$ and $T_{1/2}$ (Supplementary Fig. 1d–g). We thus conclude that the number of recycling AMPARs, but not the rate of AMPAR recycling, is reduced during LTD maintenance.

**Sorting from EE to LE is required for cerebellar LTD**. It has been reported that sorting from EE to LE is required for hippocampal LTD[12, 13], raising a possibility that the sorting is the mechanism by which number of AMPARs within the recycling loop is reduced during cerebellar LTD. Immunohistochemical analyses demonstrated that the LE marker protein Rab7 was strongly expressed in PCs (Supplementary Fig. 2a), and was specifically found in the soma, proximal dendrites, distal dendrites, and spine-like protrusions of PCs (Fig. 3a). Marker proteins of lysosomes (Lamp1) and EE (EEA1 and Rab5) were also expressed in PCs (Supplementary Fig. 2). When two marker proteins were stained together, Rab7 and Lamp1, as well as EEA1 and Rab5, were often observed to be juxtaposed in PC dendrites. Rab5 and Rab7 were also frequently found close to each other in proximal and distal dendrites. These results indicate that endosomal components exist in distal dendrites bearing dendritic spines. Thus, endosomal trafficking is likely to occur at or near PC synapses.

We next tested the involvement of EE to LE sorting in cerebellar LTD by expressing Rab7TN in PCs. Rab7 is a small GTPase regulating vesicle sorting from EE to LE[21–23], and Rab7TN has been used as a tool to block vesicle sorting in a variety of cells[24, 25], including neurons[12, 26]. To express green fluorescent protein (GFP)-fused Rab7TN (GFP-Rab7TN) or GFP alone in PCs, we injected adeno-associated virus (AAV) with a Cre recombinase-dependent genetic switch (FLEX)[27] into the cerebellar cortex of PCP2-Cre mice, which express Cre exclusively in PCs. Indeed, expression of GFP-Rab7TN was restricted to PCs (Fig. 3b) and was also observed in spines that were identified by staining with an antibody for metabotropic glutamate receptor 1 (mGluR1; Supplementary Fig. 6a). The relationship between the PF-EPSC amplitudes at the basal level and stimulus intensities in PCs expressing GFP-Rab7TN was similar to that in cells expressing GFP alone (Fig. 3c). Because expression of Rab7TN was specific to postsynaptic PCs, it is unlikely that this expression affects presynaptic release. Consistent with our speculation, the paired pulse facilitation ratio of PF-EPSCs was not significantly different between cells expressing GFP and those expressing GFP-Rab7TN (Fig. 3d). Thus, basal synaptic transmission does not appear to be affected in Rab7TN-expressing PCs. We then tested LTD in PCs expressing GFP-Rab7TN or GFP alone by monitoring PF-EPSCs. Pairing stimulation of PFs together with

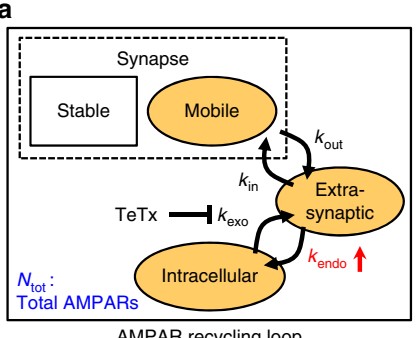

**Fig. 2** Model confirmation of the reduction in the number of recycling AMPARs. **a** Diagram of the model. Two possible pathways for implementing LTD maintenance are considered: increase in the trafficking speed in $k_{endo}$ (increased $k_{endo}$), or reduction in total AMPAR numbers in the recycling loop (reduced $N_{tot}$). **b** Changes in EPSC level (*top*), and changes in $T_{1/2}$ (*middle*) or $MR_{PF-EPSC}$ (*bottom*) of $k_{exo}$ inhibition-mediated reduction, when $N_{tot}$ is reduced (*left*) or $k_{endo}$ is increased (*right*) in the model. Values a and b were used for **c**–**e**. *Magenta dotted lines* show values calculated from experimental results shown in Fig. 1c, d. **c** Time course of changes in EPSC amplitudes, when $k_{exo}$ was inhibited in the model (*solid lines*). The *black line* was obtained by adjusting parameters (Supplementary Table 1) to reproduce the experimental result at baseline. Specific values of reduced $N_{tot}$ (a shown in **b**) and of increased $k_{endo}$ (b shown in **b**) were used to obtain *blue and red lines*, respectively. Circles are the experimental results shown in Fig. 1b. **d**, **e** $T_{1/2}$ **d** and $MR_{PF-EPSC}$ **e** in the $k_{exo}$ inhibition-dependent EPSC reductions obtained from the data shown in **c**.

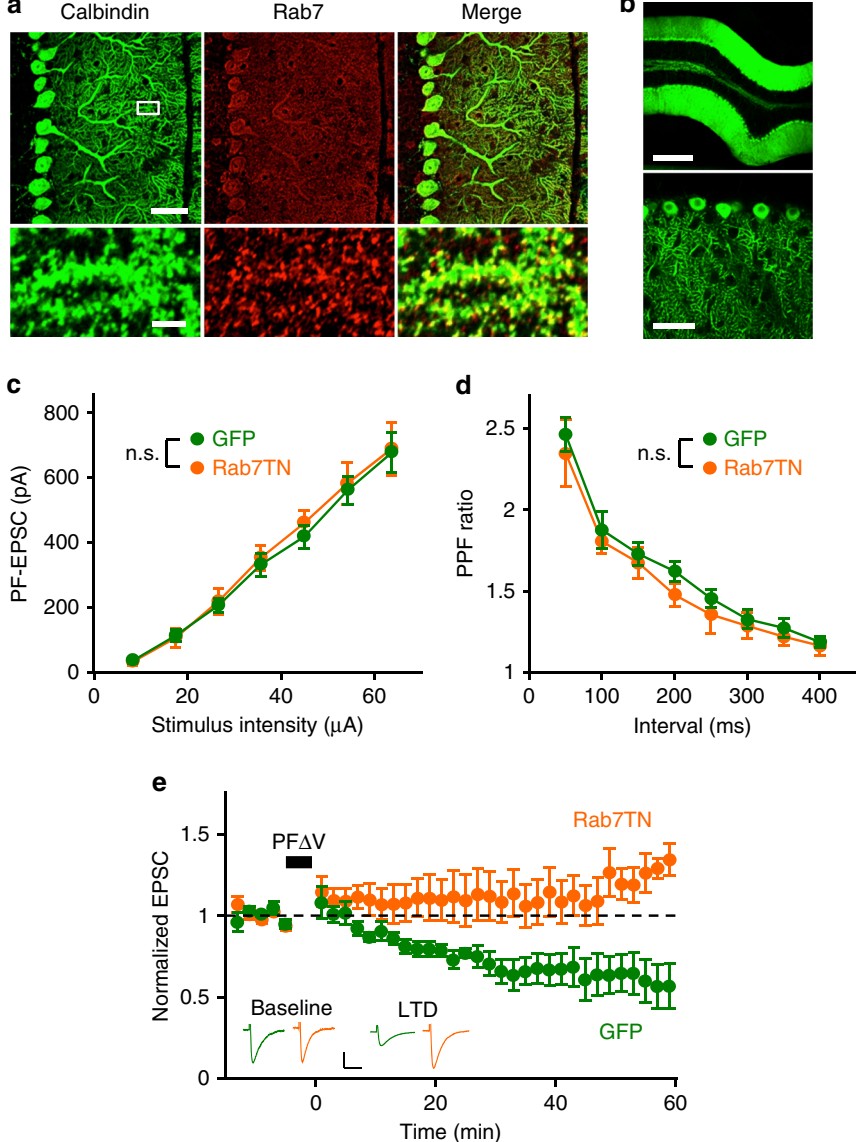

**Fig. 3** Rab7-dependent LE sorting is involved in LTD. **a** Confocal images of a cerebellar slice double stained with antibodies against calbindin (*green*) and Rab7 (*red*). Scale bars: *top*, 40 μm; *bottom*, 4 μm. The area within the *white square* is magnified in the lower panels. **b** Images of a cerebellar slice expressing GFP-Rab7TN in PCs. Scale bars: *top*, 300 μm; *bottom*, 50 μm. **c** Amplitudes of PF-EPSCs elicited by PF stimuli of increasing intensity (8.2, 17.4, 26.5, 35.6, 44.9, 54.2, and 63.6 μA), which were recorded from PCs expressing GFP alone (*green circles*, $n = 10$), or GFP-Rab7TN (*orange circles*, $n = 6$). **d** Paired pulse facilitation (*PPF*) ratios of PF-EPSCs elicited by a pair of PF stimuli with different intervals (50–400 ms) from PCs expressing GFP alone (*green circles*, $n = 13$), or GFP-Rab7TN (*orange circles*, $n = 12$). **e** Time course of LTD induced by PFΔV in PCs expressing GFP alone (*green circles*, $n = 9$), or GFP-Rab7TN (*orange circles*, $n = 8$). Averaged traces of PF-EPSCs recorded before (baseline) and after PFΔV (LTD) are shown in the inset. Calibration: 100 pA, 30 ms. PF-EPSC amplitudes shown here as well as in the subsequent figures presenting the time course of PF-EPSCs are normalized to their mean prestimulation level. n.s., not significant. Exact P values for the data sets in this figure are provided in Supplementary Table 2

PC depolarization at 1 Hz for 5 min (PFΔV) induced LTD in PCs expressing GFP, whereas it failed to induce LTD in cells expressing GFP-Rab7TN (Fig. 3e). Average amplitudes of PF-EPSCs over periods of either 5–20 min or 25–40 min after the end of PFΔV, which are defined as the expression or maintenance phase, respectively, were significantly different between PCs expressing GFP alone and those expressing GFP-Rab7TN (Fig. 5c). These results indicate that sorting from EE to LE is required for cerebellar LTD.

**Acute disruption of LE sorting reverses cerebellar LTD.**
Because the experiments using Rab7TN could not determine

when LE sorting is required following LTD induction, we developed a photosensitive Rab7TN to determine the timing of LE sorting. The blue-light sensitive LOV domain with a carboxy-terminal helical extension (Jα) derived from plant phototropin-1 has previously been used to create photoactivatable Rac1 (PA-Rac1)[14], a small GTPase regulating the cytoskeleton. We hypothesized that fusing with LOV-Jα might provide photosensitivity to Rab7 (Supplementary Fig. 3a), and first tested LOV-Jα-fused wild-type Rab7 (LOV-Rab7) in a way similar to that used to test PA-Rac1[14]. Rab7 binds to the Rab7-binding domain of Rab-interacting lysosomal protein (RBD-RILP)[28], allowing us to use a glutathione-S-transferase (GST) pull-down assay to examine binding of several constructs of LOV-Rab7

with different junctional sequences to GST-fused RBD-RILP. We found that connecting Leu546 of LOV-Jα to Arg4 of Rab7 (LOV-Rab7-546-4) resulted in a substantial reduction of the binding of Rab7 with RBD-RILP (Supplementary Fig. 3b, c). To examine the difference between the dark- and lit-state conformations, we used photoinsensitive (CA-LOV) and lit-state mimic (IE-LOV) LOV domains, as previously used for PA-Rac1. Whereas the binding of RBD-RILP to CA-LOV-Rab7-564-4 was slightly lower than that of LOV-Rab7-564-4, the binding of IE-LOV-Rab7-564-4 recovered to the level of binding of Rab7

(Supplementary Fig. 3d, e). These results suggest that fusing with LOV-Jα enables us to create a photosensitive form of Rab7, which is less active in the dark and activated by light-dependent conformational changes in LOV-Jα.

Because Rab7TN, unlike wild-type Rab7, does not bind to RBD-RILP, another assay was required to test LOV-Rab7TN. We took advantage of staining HeLa cells with LysoTracker, a dye that labels lysosomes, and tested four constructs of CA-LOV-Rab7TN or IE-LOV-Rab7TN with different junctional sequences. As previously reported[25], labeling by LysoTracker was reduced in HeLa cells expressing GFP-Rab7TN (Fig. 4a and Supplementary Fig. 4a). The intensities of GFP and LysoTracker signals were negatively correlated in cells expressing Rab7TN, but not in cells expressing GFP (Fig. 4b). The average LysoTracker intensity in cells expressing Rab7TN was significantly lower than that in cells expressing GFP alone (Fig. 4c), showing that active Rab7TN reduces LysoTracker intensity. Using these properties of Lyso-Tracker staining, we found that connecting Leu546 of LOV-Jα to Thr2 of Rab7TN (LOV-Rab7TN-546-2) resulted in an appropriately photosensitive Rab7TN molecule: the LysoTracker intensity in cells expressing IE-LOV-Rab7TN-546-2 was equivalent to that in cells expressing Rab7TN, whereas the LysoTracker intensity in cells expressing CA-LOV-Rab7TN-546-2 was equivalent to that in cells expressing GFP alone (Fig. 4 and Supplementary Fig. 4a). Although the patterns of LysoTracker labeling in cells expressing CA- or IE-LOV-Rab7TN-546-4 were similar, the effects of Rab7TN in reducing LysoTracker intensity remained even when CA-LOV-Rab7TN-546-4 was used (Fig. 4c, Supplementary Fig. 4a, b). As the expression levels of LOV-Rab7TN-546-5 and 546-6 were very weak (Supplementary Fig. 4a), we did not further analyze these constructs. We also confirmed by staining cerebellar slices with LysoTracker that expression of IE-, but not CA-, LOV-Rab7TN-546-2, reduced LysoTracker-positive puncta in PC dendrites, similar to Rab7TN (Supplementary Fig. 4c, d). These results suggest that LOV-Rab7TN-546-2 works as a photosensitive Rab7TN molecule, and from now on is referred to as PS-Rab7TN.

We then tested whether photoactivation of PS-Rab7TN truly inhibits the sorting to LE and lysosomes. It has been shown that culture cells maintained in lipoprotein-deficient medium uptake the subsequently supplied low-density lipoprotein (LDL) and the LDL is transferred into LE and lysosomes. We utilized the LDL uptake assay in HeLa cells by using DyLight550-labeled LDL (LDL-DL), and measured colocalization of LDL-DL with lysosomes that were visualized by Lamp2 staining. In HeLa cells expressing Rab7TN, LDL-DL tended to localize less in the lysosomes (Supplementary Fig. 5a). The quantification of colocalization revealed that the expression of Rab7TN significantly reduced the colocalization, compared with the expression of GFP alone (Supplementary Fig. 5b). While the colocalization

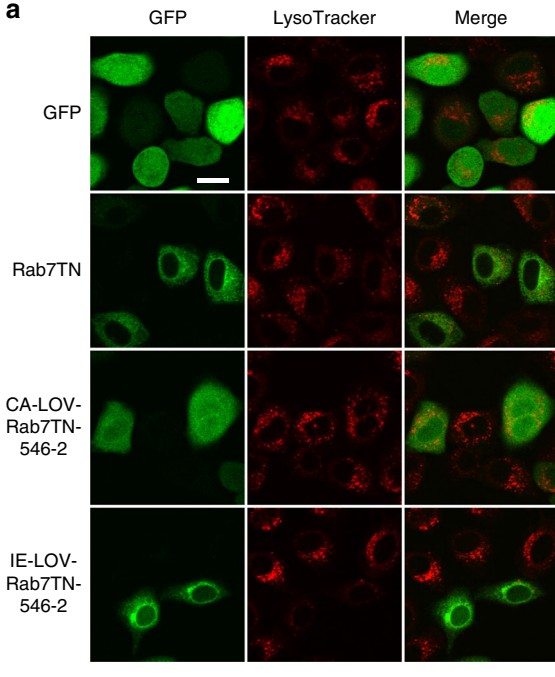

**a**

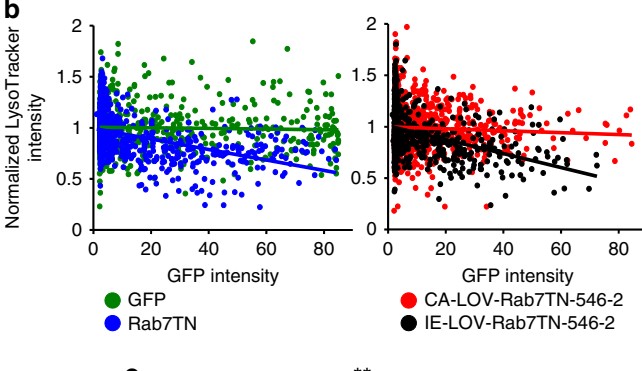

**b**

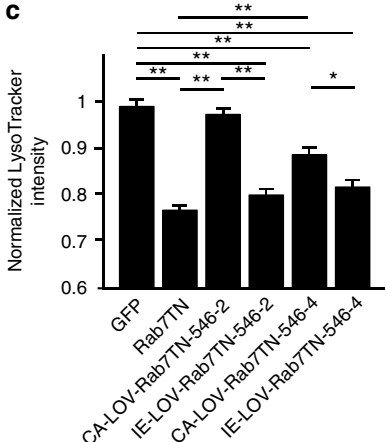

**c**

**Fig. 4** LOV-Rab7TN-546-2 appropriately works as a PS-Rab7TN. **a** Images of HeLa cells transfected with GFP, Rab7TN, CA-, or IE-LOV-fused Rab7TN-546-2 (*green*) and labeled using LysoTracker (*red*). *Scale bar*: 20 μm. **b** Relationships between GFP intensity and normalized LysoTracker intensity in individual HeLa cells transfected with GFP, Rab7TN, CA-, or IE-LOV-fused Rab7TN-546-2. Lines indicate the fit of the linear equation. Pearson's correlation coefficient: GFP, −0.035, $P = 0.33$; Rab7TN, −0.44, $P < 0.01$; CA-LOV-Rab7TN-546-2, −0.053, $P = 0.063$; IE-LOV-Rab7TN-546-2, −0.39, $P < 0.01$, ANOVA. **c** Averaged LysoTracker intensities in HeLa cells expressing GFP, Rab7TN, CA-, or IE-LOV-fused Rab7TN-546-2 or −546-4 ($n = 209$-298, GFP intensity > 10). The intensities were normalized with those of GFP-negative cells (GFP intensity < 5). *$P < 0.05$, **$P < 0.01$, one-way ANOVA followed by the Tukey's multiple comparison test. Exact $P$ values for the data sets in this figure are provided in Supplementary Table 2

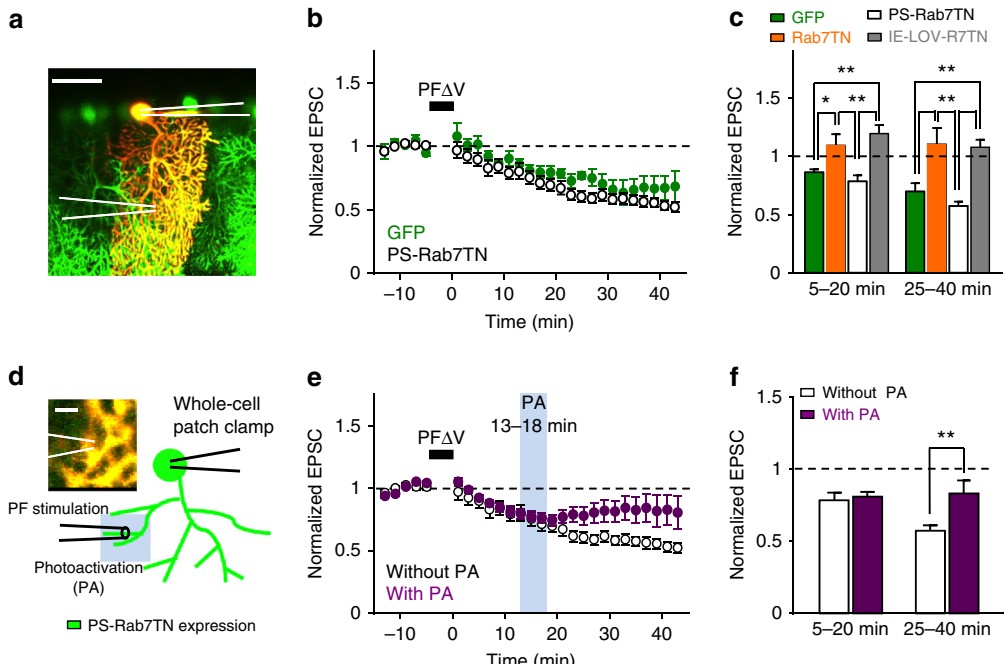

**Fig. 5** Interruption of LTD upon the photoactivation of PS-Rab7TN. **a** An image of a PC expressing PS-Rab7TN (*green*) and containing Alexa568 (*red*) introduced through a patch pipette. The positions of the patch pipette (*right*) and the stimulation electrode (*left*) are shown with *white lines*. Scale bar: 50 μm. **b** Time course of LTD induced by PFΔV in PCs expressing PS-Rab7TN without photoactivation (*open circles*, n = 10). For comparison, the time course of LTD in cells expressing GFP shown in Fig. 3e are overlaid (*green circles*). **c** Average amplitudes of normalized PF-EPSCs calculated at 5–20 min or 25–40 min after the end of PFΔV. IE-LOV-R7TN: IE-LOV-Rab7TN-546-2. **d** The design of experiments using blue laser photoactivation (*PA*). Photoactivation was applied around the PC dendrites, where the stimulation electrode is placed. Scale bar: 5 μm. **e** Time course of LTD in PCs expressing PS-Rab7TN with photoactivation at 13–18 min (*light blue column*) after the end of PFΔV (*purple circles*, n = 11). Control LTD without photoactivation shown in **b** is overlaid (*open circles*). **f** Comparison of normalized PF-EPSCs calculated at 5–20 min or 25–40 min after the end of PFΔV between with and without photoactivation at 13–18 min in PCs expressing PS-Rab7TN. *P < 0.05, **P < 0.01, two-way ANOVA followed by the Fisher's LSD test. Exact P values for the data sets in this figure are provided in Supplementary Table 2

was equivalent between cells expressing GFP alone and PS-Rab7TN, applying photoactivation onto cells expressing PS-Rab7TN caused a reduction of colocalization, which was similar to the level seen in cells expressing Rab7TN (Supplementary Fig. 5a, b). We thus confirmed that PS-Rab7TN works as a photosensitive Rab7TN that inhibits sorting toward LE and lysosomes upon photoactivation, and utilized it for the subsequent fine temporal analyses of LTD.

GFP-fused PS-Rab7TN was selectively expressed in PCs by injecting AAV into PCP2-Cre mice (Supplementary Fig. 6b). Alexa568 was included in the internal solution to visualize PCs used for PF-EPSC recordings (Fig. 5a). In cells expressing PS-Rab7TN, LTD was induced by PFΔV, and lasted as long as the slices were not exposed to photoactivation light (Fig. 5b). The average amplitudes of PF-EPSCs at either 5–20 min or 25–40 min after PFΔV were not significantly different from those in PCs expressing GFP alone, but were significantly different from those recorded from PCs expressing Rab7TN (Fig. 5c). These results confirm that PS-Rab7TN does not inhibit LE sorting without photoactivation. On the other hand, LTD was not induced in PCs expressing IE-LOV-Rab7TN-546-2 (Supplementary Fig. 7a), similar to PCs expressing Rab7TN (Fig. 5c). This implies that PS-Rab7TN expressed in PCs is able to restore the inhibitory functions of Rab7TN on LE sorting when it is fully activated by blue light.

We next tested the effects of local photoactivation with blue light (PA, 488 nm, $3 \times 10^3$ μJ, 5 min) applied over an area of 225 μm² that included dye-filled PC dendrites (Fig. 5d). Because the luminance of photoactivation used in these experiments is comparable to that used for triggering the

maximum effects of PA-Rac1[14], and is stronger than that used for interrupting LDL-DL colocalization with lysosomes in our LDL uptake assay, it is likely to be sufficient to cause the light-dependent conformational changes in LOV-Jα. In PCs expressing PS-Rab7TN, LTD was induced by PFΔV, but was prevented when PS-Rab7TN was photoactivated 13–18 min after PFΔV (Fig. 5e). The average amplitudes of PF-EPSC calculated at 25–40 min after LTD induction were significantly larger than those without photoactivation, whereas there was no significant difference in the amplitudes at 5–20 min, prior to photoactivation (Fig. 5f). In contrast, PF-EPSCs at baseline were not affected by PS-Rab7TN photoactivation (Supplementary Fig. 7b), indicating that photoactivation of PS-Rab7TN specifically affects LTD. Furthermore, LTD was normally observed even after photoactivation in PCs expressing GFP alone (Supplementary Fig. 7c), confirming that light exposure itself does not affect LTD. These results indicate that EE to LE sorting is involved in LTD around 13–18 min after LTD induction, which is approximately the time of the transition from expression to maintenance of LTD[5]. In addition, these results, together with the results showing that LTD was impaired in cells expressing Rab7TN but was intact in cells expressing PS-Rab7TN without photoactivation, confirmed that PS-Rab7TN worked as the photosensitive form of Rab7TN.

**Timing of LE sorting in cerebellar LTD.** To more precisely clarify the time window when LE sorting is required, we varied the time interval between PFΔV and photoactivation of PS-Rab7TN. Unlike the case when PS-Rab7TN was photoactivated at 13–18 min (Fig. 5e), photoactivation applied at

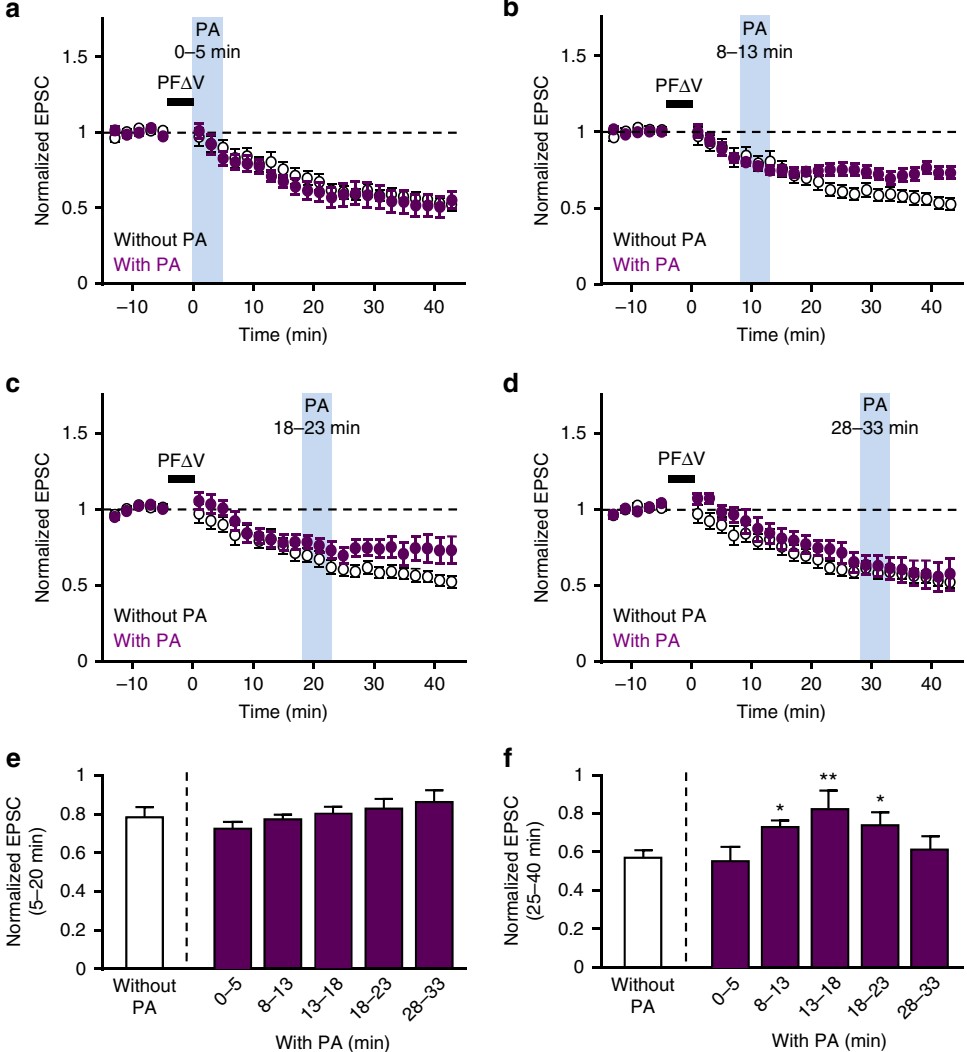

**Fig. 6** Restricted time window of the requirement of LE sorting for LTD. **a–d** Effect of photoactivation of PS-Rab7TN at various times (*light blue columns*) after PFΔV on LTD (*purple circles*; 0–5 min, $n = 9$; 8–13 min, $n = 20$; 18–23 min, $n = 11$; 28–33 min, $n = 8$). Control LTD without photoactivation shown in Fig. 5b is overlaid for comparison in each panel (*open circles*). **e, f** Average amplitudes of normalized PF-EPSCs calculated at 5–20 min **e** or 25–40 min **f** after the end of PFΔV with photoactivation at various times. *$P < 0.05$, **$P < 0.01$, two-way ANOVA followed by the Fisher's LSD test. Exact $P$ values for the data sets in this figure are provided in Supplementary Table 2

0–5 min or 28–33 min had little or no effect on LTD (Fig. 6a, d), and the amplitudes of PF-EPSC calculated both at 5–20 min and at 25–40 min were not significantly different from those without photoactivation (Fig. 6e, f). In contrast, LTD was in part impaired when the photoactivation was applied at 8–13 min or 18–23 min (Fig. 6b, c). The amplitudes of PF-EPSCs calculated at 25–40 min (Fig. 6f), but not at 5–20 min (Fig. 6e), were significantly larger than those without photoactivation. These results indicate that LE sorting is required during a defined time window, ~8–23 min after LTD induction.

Even though LTD was inhibited by photoactivation applied at three different times, the inhibition appeared to be only partial in all cases. To clarify the cause of partial inhibition, we further analyzed individual experiments when PS-Rab7TN was photoactivated at 8–23 min. For this analysis, we simplified the time course of the individual recordings by plotting PF-EPSC amplitudes measured before, during, and 15 min after photoactivation, as illustrated in Fig. 7a. We then found two patterns of responses to photoactivation at any time. In one group, PF-EPSC amplitudes returned toward the basal level after photoactivation, so that the amplitudes were larger than those during

photoactivation (Fig. 7b, *red symbols*, PF-EPSC amplitudes after photoactivation >0.8). In contrast, another group was unaffected by PS-Rab7TN photoactivation and eventually showed LTD (Fig. 7b, *blue symbols*, PF-EPSC amplitudes after the photo-activation <0.8) similar to normal. Two-way analysis of variance (ANOVA) confirmed that the former group was significantly different from the latter group and from the case without photoactivation (Supplementary Fig. 8a). On the other hand, the latter group was not distinguishable from that without photo-activation (Supplementary Fig. 8a). In addition, comparison within individual periods confirmed that the significant differ-ences arise from PF-EPSC amplitudes after the photoactivation (Supplementary Fig. 8b). Thus, we defined the former and latter groups as recovery and LTD groups, respectively. The averaged time course of LTD in each group clearly demonstrated the two distinct responses: in the recovery group, LTD was interrupted upon photoactivation and PF-EPSCs returned to their basal amplitude, whereas LTD was stably maintained in the LTD group (Fig. 7c). In the recovery group, normalized amplitudes of PF-EPSCs calculated at 35–45 min after PFΔV were ~1 and significantly larger than those without photoactivation or those in

the LTD group (Fig. 7d). Consistently, amplitudes in the LTD group were not significantly different from those without photoactivation (Fig. 7d). The presence of two distinct responses raises the possibility that there may be two types of PCs, those

showing sensitivity or insensitivity to inhibition of LE sorting. However, the calculation of recovery group fractions among all cells tested demonstrated that most of the cells belonged to the recovery group when PS-Rab7TN was photoactivated at

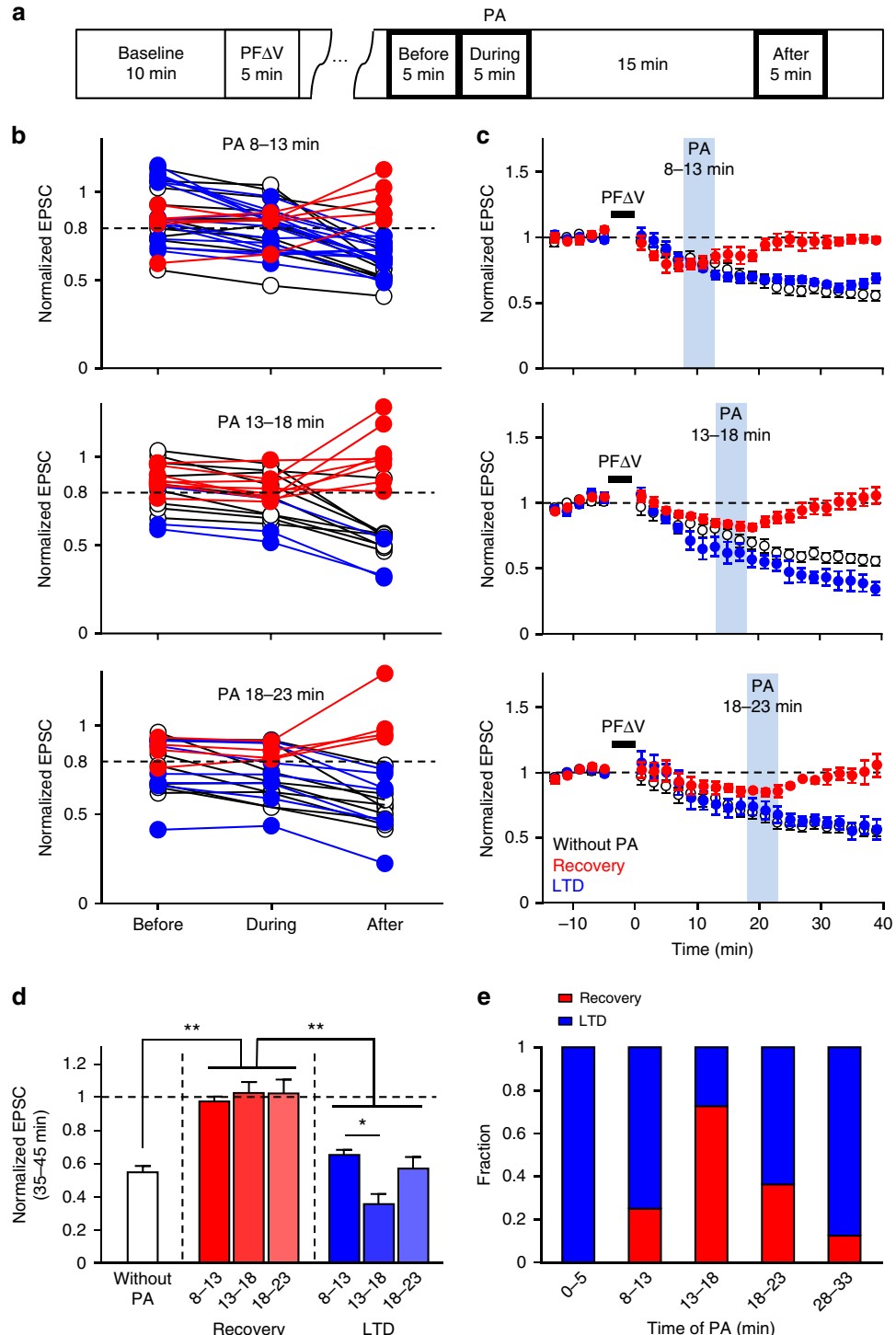

**Fig. 7** Two distinct, all-or-none responses of LTD to the transient inhibition of LE sorting. **a** Diagram demonstrating the time periods used for the calculation of the simplified time course shown in **b**. PF-EPSCs recorded during each of the three periods (before, during, and after) were averaged. **b** Amplitudes of normalized PF-EPSCs before, during, and after the photoactivtion applied at 8–13, 13–18, or 18–23 min after the end of PFΔV. Individual results showing PF-EPSC amplitudes larger or smaller than 0.8 after the photoactivation are shown in *red or blue circles*, respectively. Individual results of the experiments without photoactivation are shown in *open circles*. **c** Averaged time course of LTD in the recovery (*red circles*) or LTD (*blue circles*) group. Control LTD without photoactivation is overlaid (*open circles*). **d** Average amplitudes of normalized PF-EPSCs at 35–45 min after PFΔV in the recovery or LTD group. *$P < 0.05$, **$P < 0.01$, one-way ANOVA followed by the Tukey's multiple comparison test. **e** Fractions of the recovery (*red*) or LTD (*blue*) group among all recordings with photoactivation at various timings. Exact $P$ values for the data sets in this figure are provided in Supplementary Table 2

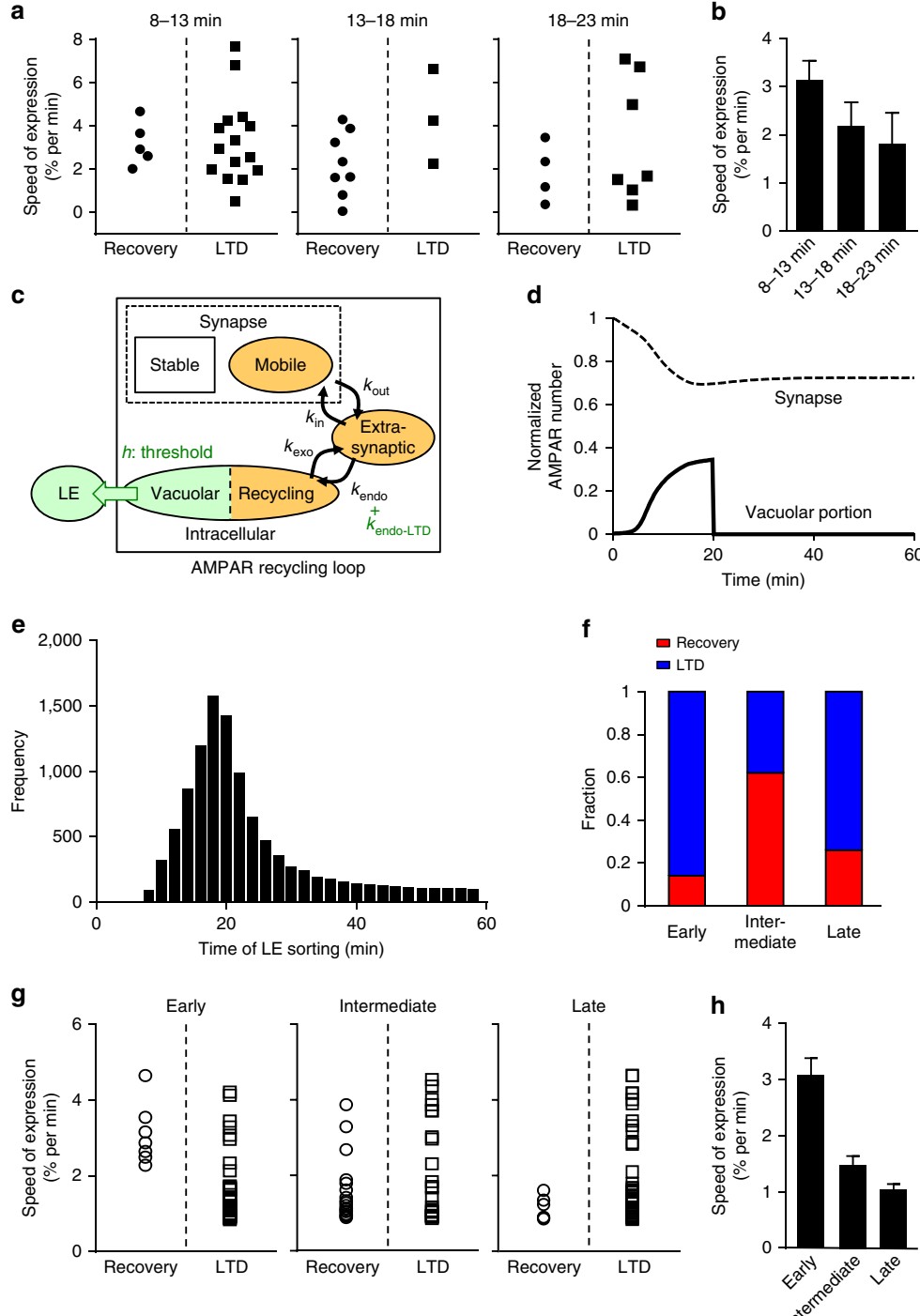

**Fig. 8** Combination of the speed of LTD expression and the threshold of LE sorting produces variability in the timing of LE sorting. **a** Experimental analyses of the speed of LTD expression (% reduction of PF-EPSCs per min) in the recovery and LTD groups upon photoactivation of PS-Rab7TN at 8–13 min (*left*), 13–18 min (*middle*), or 18–23 min (*right*). **b** Averaged speed of LTD expression in the recovery group. **c** Diagram of the advanced model. The switch-like behavior of LE sorting with a threshold (*h*) and an intracellular compartment destined for LE were added. The value of $k_{endo}$ was increased to mimic LTD expression (addition of $k_{endo-LTD}$). **d** An example of LTD (*dotted line*) and the accompanying accumulation of the vacuolar portion as well as sorting to LE of AMPARs (*solid line*) simulated by the model shown in **c**. The peak time of $k_{endo-LTD}$ ($\tau$) and threshold of LE sorting (*h*) used here were 10 and 0.345, respectively. **e** A histogram showing the appearance frequency of each timing of LE sorting, simulated by the model with varied peak time of $k_{endo-LTD}$ ($\tau$) and threshold of LE sorting (*h*). **f** Simulated fractions of recovery (*red*) and LTD (*blue*) groups. **g**, **h** Analyses of the speed of LTD expression (% reduction of PF-EPSCs per min) in the model. Individual speeds of the recovery and LTD groups **g** and the average speed of the recovery group **h** are shown. Results of the model shown in **f**–**h** are obtained from different sets of 50 examples picked by simple random sampling, among all possible examples as shown in the histogram **e**

13–18 min (72.7%), whereas a smaller fraction belonged to the recovery group in the case of 8–13 min (25%) or 18–23 min (36.4%) of photoactivation (Fig. 7e). This suggests that LTD in all cells relies on LE sorting, with the timing of sorting varying between cells. Furthermore, two distinct, all-or-none responses to the transient inhibition of LE sorting indicate that there is a threshold for triggering LE sorting during LTD. Considering the overall timing required for LTD, which is around the start of the maintenance phase, the sorting from EE to LE is likely to serve as the switch required for transitioning from the basal level to the depressed level, thereby enabling LTD maintenance.

**Determinants of the timing of LE sorting during LTD**. Our results (Fig. 7) showed that the timing of LE sorting during LTD was not constant. One possibility is that the varied speed of LTD expression results in variations in the time required to reach the threshold for LE sorting. To test this possibility, the speed of LTD expression was compared between recovery and LTD groups upon photoactivation at different times (Fig. 8a). Linear function ($f(t) = -\alpha t + \beta$) was used to fit time course of normalized PF-EPSCs at the period of first 8 min after PFΔV, and the magnitude of slope ($\alpha$) was used as the speed of LTD expression. Even when the speed of expression was similar, LTD could fall into either the recovery or LTD group with any time of photoactivation, indicating that the timing of LE sorting does not completely depend on the speed of LTD expression. However, the speed of LTD expression in the recovery group tended to correlate with the timing of photoactivation: faster LTD tended to occur with early photoactivation (Fig. 8b). These results suggest that the timing of LE sorting is partially determined by the speed of LTD expression, and may rely on the combination of the speed of LTD expression and another factor.

To further investigate the cause of the varied timing of LE sorting, we advanced the model by adding a process reflecting accumulation and LE sorting of AMPARs (Fig. 8c). This addition was based on the properties of conversion from Rab5 to Rab7 on endosomes. The Rab5-Rab7 conversion is required for and accompanied with sorting from EE to LE, and exhibits threshold behavior: Rab5 gradually accumulated onto EE suddenly disappears, to be replaced by Rab7 during the process of EE becoming LE[29]. Therefore, the dynamics of Rab5, accumulation and sudden disappearance on EE, could be utilized to describe the accumulation and LE sorting of AMPARs. We also added a portion in the intracellular pool destined for LE (Fig. 8c), corresponding to the vacuolar portion of EE defined previously[30, 31]. Figure 8d shows an example of the time course of LTD (dotted line) triggered by the enhancement of internalization (addition of $k_{endo\text{-}LTD}$ shown in Fig. 8c), shown as a piecewise-defined concave function (Methods section, Supplementary Fig. 9a). The accumulation of AMPARs within the vacuolar pool (solid line in Fig. 8d) was accompanied by gradual expression of LTD, and a sudden decrease in AMPAR numbers in the vacuolar portion represents sorting to LE. Thus, the revised model could reproduce both the kinetics of LTD as well as the inferred timing of LE sorting.

In addition to the speed of LTD expression, having a threshold for LE sorting could determine the timing of LE sorting, according to the dynamics of the accumulation of AMPARs within the vacuolar pool. We thus changed two parameters, the peak time of $k_{endo\text{-}LTD}$ ($\tau$), which controls the speed of LTD expression (Supplementary Fig. 9b), and the threshold of LE sorting ($h$) in the model. In fact, altering either $\tau$ or $h$ was capable of varying the timing of LE sorting (Supplementary Fig. 9c, d). We then tested whether combinations of different values for $\tau$ and $h$ optimally reproduced our experimental results. The

combinations of different $\tau$ and $h$ values resulted in varied timing for LE sorting with a non-uniform distribution (Fig. 8e). Fifty examples were randomly chosen and were categorized into the recovery or LTD group under the assumption that the recovery group consists of examples with LE sorting occurring during early, intermediate, or late inhibition of LE sorting. We confirmed that the recovery group accounted for only a small fraction of the early or late inhibition, but accounted for a large fraction of the intermediate inhibition (Fig. 8f), consistent with the experimental results shown in Fig. 7e. We also analyzed the speed of LTD expression ($\alpha$) in the recovery and LTD groups using 50 randomly picked examples, by using a similar method to calculate the speed in experimental results. The results exhibited two features of the experimental results (see Fig. 8a, b): (1) there was an overlapping range in speed of LTD expression between recovery and LTD groups (Fig. 8g and Supplementary Fig. 10a-c), (2) and the speed of LTD expression in the recovery group correlated with the timing of inhibition (Fig. 8h and Supplementary Fig. 10d). These analyses using our model suggest that the timing of LE sorting during LTD is determined by a combination of the speed of LTD expression and the threshold for LE sorting.

## Discussion

Regulation of AMPAR trafficking plays crucial roles in many forms of long-term synaptic plasticity. The expression of cerebellar LTD also relies on AMPAR internalization[8]. However, the events that follow AMPAR internalization, as well as the mechanism leading to stable LTD, has remained unclear. We demonstrated by experiments using TeTx, together with computational modeling, that the number of recycling AMPARs is reduced during the maintenance of LTD. Our newly developed PS-Rab7TN revealed that transient sorting from EE to LE is required around the time when depression of synaptic transmission reaches a maximum level. When sorting to LE was blocked at the appropriate time by photoactivation of PS-Rab7TN, PF-EPSC amplitudes returned to basal levels, indicating that the sorting of AMPARs to LE is an all-or-none switch system leading to stable depression. Furthermore, our advanced model revealed that the timing of LE sorting during LTD varies because of variable speed of LTD expression and a threshold for LE sorting. Thus, our study clarified the timing and properties behind transition from expression to maintenance of LTD.

When we tested LTD in PCs expressing Rab7TN, LTD was completely abolished. Because constitutive expression of Rab7TN might cause secondary effects on endosomal trafficking of AMPARs, similar to the increased recycling of nutrient transporters[32], these experiments could not elucidate precise timing of LE sorting leading to LTD. In contrast, experiments using PS-Rab7TN enabled us to transiently inhibit LE sorting. We then found that LE sorting as a whole works around 8–23 min after the end of LTD stimulation. Although the gradual expression of LTD starts almost immediately after the end of a stimulus, LTD was not affected by inhibition of LE sorting at 0–5 min and was blocked only in a fraction of PCs (25%) at 8–13 min. These results suggest that LE sorting is not directly involved in the initial expression of LTD, and support an idea that the lack of transient depression in PCs expressing Rab7TN is due to an indirect effect of constitutive Rab7TN expression.

In general, internalized membrane molecules delivered to EE are either recycled back to the plasma membrane or sorted to LE, and the separation is due to specific targeting information encoded within the molecules destined for LE[30]. Whereas AMPARs are constitutively recycled back to synapses during basal synaptic transmission, internalized AMPARs are sorted into LE during LTD, indicating that they must contain targeting

information. Even though we have shown the inhibition of LTD by the photoactivation of PS-Rab7TN, molecules other than Rab7 likely convey the targeting information, considering that Rab7 generally regulates vesicle sorting. In hippocampal LTD, the transport of AMPARs from EE to LE requires binding of the dephosphorylated form of the AMPAR auxiliary protein stargazin with adaptor protein 3 (AP-3), which is a molecule targeted to LE[13]. Because dephosphorylation of stargazin is also required for cerebellar LTD[33], this tripartite complex of AMPARs, stargazin, and AP-3 may similarly work in LE sorting of internalized AMPARs during cerebellar LTD. Alternatively, the protein interacting with C kinase 1 (PICK1) might mediate the interaction of AMPARs and AP-3, considering reports showing the binding of PICK1 with AMPARs during LTD[34] and capability of PICK1 to bind with AP-3[35]. If stargazin or PICK1 mediate the LE sorting of AMPARs during cerebellar LTD, they would be also transported to the LE together with AMPARs. In the present study, we showed that PF-EPSCs recovered to the basal level when LTD was blocked by the photoactivation of PS-Rab7TN, suggesting that internalized AMPARs originally destined for LE, could be quickly redirected to the recycling pathway if LE sorting does not occur at the proper time. Thus, the targeting information on internalized AMPARs during LTD probably relies on the temporary modulation of AMPARs or binding partners during initial phase of LTD. Although Rab7 also regulates the retromer[36], which mediates cellular trafficking from endosomes to the trans-Golgi network and is involved in AMPAR trafficking[37,38], it is unlikely that LTD was inhibited due to photoactivation of PS-Rab7TN interrupting retromer, because retromer seems to regulate an increase in synaptic AMPAR numbers, LTP[39].

A previous study suggested that a system working downstream of the PKC-MAPK positive feedback loop has a threshold to produce LTD[11]. The current study demonstrated that timely applied photoactivation of PS-Rab7TN resulted in two distinct responses, either normal LTD or recovery of transmission back to the basal level. Thus, LE sorting is likely to be the system that exhibits threshold behavior that determines whether LTD can be reversed or sustained. Considering the reversible nature of the LOV-Jα domain[14], the inhibition of LE sorting by the photoactivation of PS-Rab7TN is not expected to last very long; however, LTD could be blocked by 5 min of photoactivation. This suggests that LE sorting occurs only for a short time, on the order of 5 min or less. This transient property of LE sorting during LTD seems to comply with its function as a switch required for the transition from LTD expression to maintenance. In addition, we included the threshold and switch-like behavior of the Rab5-Rab7 conversion[29,40] in our model, and this allowed the model to reproduce several aspects observed in the experimental results, leading to the conclusion that LE sorting works as a switch to accomplish the prompt transition to LTD maintenance following accumulation of internalized AMPARs.

A comparison of our model predictions with our experimental results suggests that the timing of LE sorting depends on a combination of two factors: the speed of LTD expression and the threshold of LE sorting. Different time courses of LTD expression have been observed depending on the type of stimulus and stimulus intensity[3,5]. Thus, the timing of LE sorting may vary according to the type of stimuli. However, various speed of LTD expression was observed by using a single protocol (PFΔV) in the present study. The kinetic model predicted that the stimulus-dependent variability of LTD results from the efficacy of stimulus in activating the PKC-MAPK positive feedback loop[6]. Because many molecules are involved in activation of this loop, the efficacy may rely on the number of available signaling molecules around the stimulated synapses. In addition, the efficacy may also vary depending on the stochastic nature of signaling molecules[41].

Thus, the number or the stochastic nature of molecules may alter the timing of LE sorting. It is completely unknown as to how the threshold of LE sorting is determined in PCs. However, because the threshold can be interpreted as the capacity of EE to accept AMPARs during gradual internalization, the variability in threshold suggests the non-uniform capacity of EE in PCs. Properties of LE sorting that occur after the gradual accumulation of AMPARs in the vacuolar portions of EE might be beneficial for the robustness of LTD, despite variable conditions around stimulated synapses in PCs. Furthermore, the variable conditions we suggest may imply multiple states of synapses, which are proposed by the cascade model to be required for the memory storage with long retention times[42].

Several studies using cultured PCs have demonstrated the requirement of transcriptional and translational events for maintenance of the late phase of LTD, which is defined as the phase beginning ~60 min after induction[43,44]. The present study showed that maintenance at least initially relies on a reduction in the number of recycling AMPARs. These two results taken together suggest that maintenance of the decreased number of recycling AMPARs and the consequent reduction in the number of synaptic AMPARs for an extended length of time may require the remodeling of synaptic molecules, as described in the remodeling of postsynaptic density molecules in hippocampal CA1 pyramidal neurons[45]. Another study using cultured PCs reported the requirement of dynamin-dependent endocytosis in the late phase of LTD maintenance, but not in the basal level[46], whereas we found that AMPAR internalization was not enhanced during the early phase of LTD maintenance. A possible explanation to reconcile these results is that the molecular mechanisms, rather than the rate, of AMPAR recycling may be altered as a part of remodeling of synaptic molecules in the late phase of LTD maintenance. Although the remodeling may be required for the late phase of LTD maintenance, transient activation of LE sorting at the end of enhanced AMPAR internalization and the resultant reduction in recycling AMPAR numbers appear to be the origin of the early phase of LTD maintenance.

Because sorting from EE to LE regulates the composition of many membrane proteins, this sorting is presumably involved in many dynamic neuronal functions. The importance of dynamic functions should be more readily demonstrated in studies using optogenetic control. Using our newly developed PS-Rab7TN, we found that LE sorting works as a switch to accomplish the transition into the stable maintenance of LTD, which is distinct from the stable basal state. This implies that LE sorting can be considered as a bistable switch. Because it was theoretically proposed that stable LTD requires a bistable switch working downstream of another bistable switch of the PKC-MAPK positive feedback loop[47], LE sorting appears to satisfy the conditions required for stable LTD. Further, considering that LE sorting is required for hippocampal LTD[12,13] presumably after transient increase in AMPAR endocytosis[48,49], it may similarly work as a switch required for the stable maintenance of hippocampal LTD as well as other forms of LTD arising from the reduced number of synaptic AMPARs. In general, an important feature of long-term synaptic plasticity is sustained changes in synaptic transmission. Our study demonstrating switch functions of LE sorting for cerebellar LTD provides a concept that a switch-like system is appropriate for the transition from the basal state of synaptic transmission into the maintenance phase of synaptic plasticity.

## Methods

**Mice**. All procedures involving mice were performed according to the guidelines of the Institutional Animal Care and Use Committee of Korea Institute of Science and Technology. C57BL/6, ICR, and PCP2-Cre transgenic mice (Jackson Laboratories,

B6.129-Tg(Pcp2-cre)2Mpin/J) were used in this study. Heterozygous PCP2-Cre mice were obtained by crossing male PCP2-Cre mice with female ICR mice.

**Patch-clamp recording and photoactivation.** Chemicals were obtained from Sigma or Wako Pure Chemical Industries unless otherwise specified. Fresh sagittal slices of the cerebellum were prepared from 18- to 25-day-old mice of either sex. Slices were bathed in artificial cerebrospinal fluid (ACSF) containing (in mM): 125 NaCl, 2.5 KCl, 1.3 MgCl$_2$, 2 CaCl$_2$, 1.25 NaH$_2$PO$_4$, 26 NaHCO$_3$, 20 glucose, and 0.01 bicuculline methochloride (Tocris Bioscience). Patch pipettes (resistance 5–9 MΩ) were filled with (in mM): 130 potassium gluconate, 2 NaCl, 4 MgCl$_2$, 4 Na$_2$-ATP, 0.4 Na-GTP, 20 HEPES (pH 7.2), and 0.25 EGTA. For the photo-activation experiments, Alexa568 (0.125 mM, Thermo Fisher Scientific) was added to the internal solution to visualize the recorded PCs. For blocking exocytosis, 200 nM TeTx (List Biological Laboratories) along with Alexa568 was included in the internal solution. When GFP-conjugated proteins were virally expressed, whole-cell patch-clamp recordings were made from PCs expressing GFP, which were identified by their GFP fluorescence under a microscope (Olympus BX61WI or Nikon FN1). Imaging of GFP or Alexa568 was performed using a confocal microscope (Olympus FV1000).

Whole-cell patch-clamp recordings were made from PCs in cerebellar sagittal slices. PF-EPSCs were recorded and LTD stimulation was applied as described previously[16]. Briefly, PFs were activated to evoke PF-EPSCs in PCs (holding potential of −70 mV) with a glass stimulating electrode on the surface of the molecular layer. PF-EPSCs were acquired and analyzed using pClamp software (Molecular Devices). To evoke LTD by electrical stimulation, PF stimuli were paired with PC depolarization (0 mV, 200 ms) 300 times at 1 Hz. This protocol was previously used to identify the time when the PKC-MAPK positive feedback loop works for LTD[5], and was used also in the current study to trigger similar time course of LTD. Data were accepted if the series resistance changed by <30%, the input resistance was >70 MΩ, and the holding current changed by <20%. The criteria were pre-established. To analyze the effects of TeTx on PF-EPSCs during LTD maintenance, slices were treated with K-glu for 5 min, and then kept in normal ACSF for 30 min before being transferred to the recording chamber. It usually took us about 5 min to make whole-cell patch clamp, and PF-EPSC amplitudes started to decrease within 6–8 min after establishing the whole-cell configuration (Supplementary Fig. 1a), as previously reported[15]. Thus, the effects of TeTx on PF-EPSCs were tested around 40 min after K-glu treatment.

For the analyses of TeTx-dependent reduction of PF-EPSCs, the time to reach half-maximum reduction ($T_{1/2} = (\ln 2)/k$) and the maximum reduction in PF-EPSCs ($MR_{PF-EPSC} = 1 − f(\infty) = 1 − B$) were extracted by fitting an exponential decay function ($f(t) = Ae^{-kt} + B$). To obtain the speed of LTD expression ($\alpha$), a linear function ($f(t) = −\alpha t + \beta$) was used to fit time course of normalized PF-EPSCs for the first 8 min after PFΔV, which do not include effects of photoactivation even when applied at 8–13 min, and the magnitude of slopes ($\alpha$) were used as the speed of LTD expression (% reduction of PF-EPSCs per min).

For the photoactivation of PS-Rab7TN in PCs, a scanning laser in the Olympus FV1000 microscope was used. Blue light (488 nm, 10 μW) was delivered to the slice via a ×60 water-immersion objective. Photoactivation was accomplished with continuous scanning of a 15 × 15 μm$^2$ region of interest at a speed of 10 μs per pixel for 5 min. The light spot was focused on the dendrites of PCs, where PFs were electrically stimulated.

**Development of PS-Rab7TN.** To find an appropriate construct of PS-Rab7TN, we tested GFP-fused LOV-Rab7TN with different junctional sequences by Lyso-Tracker staining in HeLa cells. complementary DNA encoding the LOV2 domain of phototropin-1 (404–546), including the C-terminal helical extension (Jα), was obtained by PCR from the pTriEx-mCherry-PA-Rac1 plasmid (Addgene)[14], and the amplified fragments were used for megaprimer PCR-based mutagenesis to create the construct of GFP-fused LOV-Rab7-546-2. Truncations/extensions of the LOV2-Jα C terminus (545–547) or the Rab7 N terminus (2–7) was generated by site-directed mutagenesis from this construct. Site-directed mutagenesis was also used to add mutations to Rab7TN as well as CA- and IE-LOV.

The pull-down assay with GST-fused RBD-RILP was performed as described previously[50] with slight modifications, to test the binding between RILP and several constructs of LOV-Rab7 with different junctional sequences. In brief, GST-fused RBD-RILP expressed in *Escherichia coli* BL21 was bound to glutathione Sepharose (GE Healthcare). Plasmids were transfected into HEK293T cells by transient transfection using Lipofectamine 2000 (Thermo Fisher Scientific). The cells were then lysed in pull-down buffer containing (in mM): 20 HEPES (pH 7.4), 100 NaCl, 5 MgCl$_2$, 1% Triton-X 100, and 1× protease inhibitor mixture (Nacalai Tesque). After the centrifugation, supernatants including 200 μg of protein were added to the GST-fused RBD-RILP (60 μg) bound to glutathione Sepharose. The Sepharose beads were incubated for 1 h at 4 °C and then washed with pull-down buffer. The bound proteins were eluted by sodium dodecyl sulfate polyacrylamide gel electrophoresis sample buffer and subjected to immunoblotting analyses using an anti-GFP (Roche Applied Science, 11814460001) and horseradish peroxidase-conjugated anti-mouse IgG antibodies (GE Healthcare). For quantification, band intensities were measured using ImageJ software (National Institutes of Health, http://imagej.nih.gov/ij/), and ratios of band intensities in the pull-down samples to those in the input samples were normalized with the ratios of GFP-Rab7.

For the assay using LysoTracker staining in HeLa cells (from the Korean Cell Line Bank), plasmids were transfected into HeLa cells by transient transfection using Lipofectamine 2000. At 24 h after transfection, cells were incubated in culture medium containing 50 nM LysoTracker Deep Red (Thermo Fisher Scientific) for 30 min, and then the medium was replaced with extracellular solution containing (in mM): 5 HEPES (pH 7.4), 150 NaCl, 4 KCl, 2 CaCl$_2$, 1 MgCl$_2$, and 5 glucose. Images were taken using a confocal microscope (Olympus FV1000). Intensities of LysoTracker and GFP signals in individual cells were measured using ImageJ software, and intensities of LysoTracker signals were normalized with those of GFP-negative cells (GFP intensity <5). To stain cerebellar slices with LysoTracker[51], fresh slices were obtained from PCP2-Cre mice subjected to the stereotaxic injection of AAV, and were incubated in the ACSF including 200 nM LysoTracker Deep Red for 20 min. Images of GFP and LysoTracker signals in PC dendrites were taken using a confocal microscope (Olympus FV1000). Numbers of LysoTracker-positive puncta were counted in GFP-positive PC dendrites using ImageJ software.

For LDL uptake assay, HeLa cells were incubated in culture medium supplemented with lipoprotein-deficient serum (Millipore) for 18–22 h. LDL-DL (Cayman, 20 μg ml$^{-1}$) were added in the culture medium for 30 min. After washing with PBS, HeLa cells were incubated in the medium without LDL-DL for another 30 min, and then were fixed by paraformaldehyde to proceed staining with Lamp2 antibody (abcam, ab25631). For the photoactivation, blue light was applied onto cells during whole 60 min incubation by using LED array light sources (peak at 470 nm, ThorLabs) placed on culture plates. Considering the illumination intensity (13 mW by 70 mm$^2$ sensor) of LED, the energy used for the photoactivation can be calculated as 0.7 μJ μm$^{-2}$, which is 20 times lower than that of the scanning laser. Localization of LDL-DL in Lamp2-positive compartment was quantified by previously described colocalization analysis[52] with slight modification. In brief, images from 7 to 10 different fields (211.7 × 211.7 μm$^2$) were taken using a confocal microscope (Nikon) in one experiment. Individual cells expressing GFP, GFP-fused Rab7TN, or PS-Rab7TN were identified by GFP signals. Colocalization areas of LDL-DL and Lamp2 were established when pixels included both signals over a threshold that was determined by automatic thresholding of ImageJ software. Intensities of LDL-DL signals in colocalization areas were divided by intensities of total LDL-DL signals in individual cells, and the ratios were used as LDL-DL in Lamp2-positive compartments (% of total LDL-DL). Total numbers of cells measured were 117–160 in 3–4 separate experiments per different conditions.

**Immunohistochemistry.** Primary antibodies used were mouse anti-Rab7 (Sigma, R8779), rabbit anti-Lamp1 (abcam, ab24170), rabbit anti-Rab5 (abcam, ab13253), mouse anti-EEA1 (Sigma, E7659), guinea pig anti-calbindin (Synaptic Systems, 214 004), and mouse anti-mGluR1 (BD Biosciences, 610964). Secondary antibodies used were Alexa Fluor 647-conjugated anti-rabbit or anti-mouse IgG (Thermo Fisher Scientific, A-21245 or A-21236), Alexa Fluor 488-conjugated anti-rabbit or anti-mouse IgG (Thermo Fisher Scientific, A-11034 or A-11001), and DyLight 405-conjugated anti guinea pig IgG (Jackson ImmunoResearch, 706-475-148).

For immunohistochemistry of cerebellar slices, mice were anesthetized and perfused transcardially with 4% paraformaldehyde in 0.1 M sodium phosphate buffer (pH 7.4). Cerebella were postfixed overnight at 4 °C followed by sectioning (40 μm) using a vibrating microtome (Leica VT 1200 S). The sections were blocked in 5% normal goat serum in phosphate-buffered saline (PBS), incubated in primary antibodies overnight at 4 °C, washed several times, and then incubated in secondary antibodies for 3 h at room temperature. Single optical sections were acquired using an A1R laser scanning confocal microscope (Nikon).

**Virus preparation and stereotaxic injection.** A plasmid for AAV vectors with the short synapsin promoter and FLEX switch cassette was used for the expression of GFP or several forms of GFP-fused Rab7TN in PCs. AAV vectors were produced according to the protocol provided by the Salk Institute viral vector core facility (http://vectorcore.salk.edu/index.php) with slight modifications. In brief, HEK293T cells were cotransfected with a mixture of three plasmids, one AAV plasmid, the pHelper plasmid (Agilent Technologies), and the serotype 1 plasmid (Vector Core of the University of Pennsylvania), using calcium phosphate. At 72 h after transfection, AAV vectors were harvested by sonication. Then, AAV vectors were collected from the cell lysate by gradient ultracentrifugation using a Beckman NVT90 rotor at 183,000×g for 47 min, dialyzed in PBS with D-sorbitol, con-centrated by centrifugal filter devices (Millipore), and stored at −80 °C. The AAV titer was estimated by quantitative PCR of DNase-I-treated AAV.

AAV vectors were injected into the cerebellum of 8- to 11-day-old heterozygous PCP2-Cre mice of both sexes by the procedure previously described[53]. Briefly, mice were anesthetized by Avertin (250 μg per gram body weight) and mounted on a stereotaxic stage. The cranium over the cerebellar vermis was exposed by a midline sagittal incision. A hole was made ~2.5–3 mm caudal from lambda by the tip of forceps, and a glass needle was placed in lobe IV–V of the cerebellar vermis. The virus solution (1 μl total) was injected at three different depths using a Nanoliter 2010 injector (World Precision Instruments). After the surgery, mice were kept on a heating pad until they recovered from the anesthesia and then they were returned to their home cages. Mice subjected to AAV injection were used for the following experiments at 7 to 14 days after the injection.

**Statistical analyses of the experimental data.** Statistical differences were determined by the unpaired Student's $t$-test (for two-group comparisons) and one-way or two-way ANOVA followed by the uncorrected Fisher's least significant difference (LSD) test or the Tukey's multiple comparison test (for more than two-group comparisons or for comparing relationships) for approximately normally distributed data with similar variances. However, the individual data were analyzed in case that the data set appeared to contain multi-modality as we showed in Fig. 7. The correlation between GFP and LysoTracker signals in HeLa cells was tested using Pearson's correlation coefficient. Sample sizes were not statistically pre-determined but conform to similar studies. The experiments were not randomized. The investigator was blinded for all analysis and quantifications. All data are presented as

mean ± s.e.m. Analyses were performed using MATLAB, GraphPad Prism 6 and OriginPro softwares. $P$ values and statistical test used are summarized in Supplementary Table 2.

**Construction of the model for synaptic AMPAR trafficking.** We built the model by reference to models of AMPAR trafficking previously reported[54, 55]. The model consists of four AMPAR pools: a mobile synaptic pool ($M$) that recycles, a stable synaptic pool ($C$) that does not undergo dynamic recycling, as well as extrasynaptic ($E$), and intracellular ($I$) pools. To adjust with the observation of experiments using TeTx, which show remaining PF-EPSCs even after the blockage of exocytosis by TeTx, the synaptic pool ($S$) was divided into 2 pools, $C$ and $M$, and no receptor exchanges were assumed between these pools in the time scale dealt with in this study (1–2 h). Thus, AMPARs were considered to be recycled through three pools, $M$, $E$, and $I$, and AMPAR trafficking in this system was formulated as a set of linear differential equations, as follows,

$$\frac{\mathrm{d}[M]}{\mathrm{d}t} = -k_{\mathrm{out}}[M] + k_{\mathrm{in}}[E] \tag{1}$$

$$\frac{\mathrm{d}[E]}{\mathrm{d}t} = -(k_{\mathrm{endo}} + k_{\mathrm{in}})[E] + k_{\mathrm{out}}[M] + k_{\mathrm{exo}}[I] \tag{2}$$

$$\frac{\mathrm{d}[I]}{\mathrm{d}t} = -k_{\mathrm{exo}}[I] + k_{\mathrm{endo}}[E] \tag{3}$$

where $k_{\mathrm{in}}$ and $k_{\mathrm{out}}$ are the rate constants of transition between the extrasynaptic area and synaptic mobile portion, and $k_{\mathrm{endo}}$ and $k_{\mathrm{exo}}$ are the rate constants of internalization by endocytosis and insertion by exocytosis, respectively. The square brackets in the equations represent the number of AMPARs in each component, which are considered to be relative values. The number of AMPARs in a synapse ($[S] = [C] + [M]$) can be calculated by the solution of the equations, and the nor-malized value $[S]/[S]_0$, where $[S]_0$ stands for the initial synaptic AMPAR number before TeTx treatment, was used as the normalized EPSC level. TeTx treatment was mimicked by changing the $k_{\mathrm{exo}}$ value to a small value (0.03). Parameters ($k_{\mathrm{in}}$, $k_{\mathrm{out}}$, $k_{\mathrm{endo}}$, and $k_{\mathrm{exo}}$) were determined by fitting to the experimental results of TeTx-dependent PF-EPSC reduction at baseline, and are shown in Supplementary Table 1. To determine the initial values of kinetics ($k$), the ratio of $k_{\mathrm{in}}/k_{\mathrm{out}}$ and $k_{\mathrm{endo}}/k_{\mathrm{exo}}$ were defined as constant, because of the stable baseline. In addition, $k_{\mathrm{in}}$ and $k_{\mathrm{out}}$ were assumed to be larger than $k_{\mathrm{endo}}$ and $k_{\mathrm{exo}}$, considering that diffusion would be faster than endocytosis or exocytosis. Two conditions possibly imple-menting LTD maintenance, namely, reduced total AMPAR numbers ($N_{\mathrm{tot}} = [S] + [E] + [I]$) or increased $k_{\mathrm{endo}}$, were tested, and the ratios of resultant $[S]$ to initial synaptic AMPAR numbers without altering $N_{\mathrm{tot}}$ or $k_{\mathrm{endo}}$, were used as changes in EPSC, to examine whether altering $N_{\mathrm{tot}}$ or $k_{\mathrm{endo}}$ results in a reduction in EPSC similarly to LTD (Fig. 2b, top panels). Changes in $T_{1/2}$ (Fig. 2b, middle panels) and $MR_{\mathrm{PF-EPSC}}$ (Fig. 2b, bottom panels) of $k_{\mathrm{exo}}$ inhibition-mediated EPSC reduction under conditions of reduced $N_{\mathrm{tot}}$ or increased $k_{\mathrm{endo}}$ are presented as ratios of $T_{1/2}$ and $MR_{\mathrm{PF-EPSC}}$ to those without altering $N_{\mathrm{tot}}$ or $k_{\mathrm{endo}}$.

To take into account the possibility of gradual exocytosis blockade, the strategy to mimic TeTx treatment was modified into the exponentially decaying $k_{\mathrm{exo}}$ with the lowest value of 0.03 and the half decay period of 5 min (Supplementary Fig. 1b). To reproduce the experimental results by the modified model, we scanned $k_{\mathrm{endo}}$ values, and found a value (0.55 min$^{-1}$) providing the best fit by evaluating the sum of squared residuals (Supplementary Fig. 1c). Other parameters and equations remained the same as described above. The basal AMPAR endocytosis rate in the modified model was estimated from the time course of EPSC reduction when the $k_{\mathrm{exo}}$ value was suddenly reduced to a small value (0.03).

All procedures were scripted in MATLAB software. Note that we did not consider the secretory pathway, which would supply newly synthesized AMPARs through endoplasmic reticulum and Golgi apparatus, because secretory pathway is presumably very slow process in comparison with the time scale we handled, considering a few days of half-life of AMPAR subunit proteins[56–58]. Indeed, including slow secretory pathway (a half-time of 47 h) in our model had little or no effect on the EPSC reduction.

**Model modification to characterize LE sorting during LTD.** The model was modified by adding three factors required to characterize LE sorting during LTD.

First, in addition to four AMPAR pools, we added a vacuolar portion ($V$) to the intracellular pool, which is destined to become LE[30, 31]. Second, to mimic the enhancement of AMPAR endocytosis, which is responsible for LTD expression, $k_{\mathrm{endo-LTD}}$ was included during the time of LTD expression ($k_{\mathrm{endo}} + k_{\mathrm{endo-LTD}}$). The $k_{\mathrm{endo-LTD}}$ was described by a piecewise-defined concave function, which consists of a Gaussian rising ($0 \le t < \tau-3$), a steady value ($\tau-3 \le t < \tau + 3$), and an exponential decay with a time constant of $\tau + 3$ ($\tau + 3 \le t$), where $\tau$ is the peak time of $k_{\mathrm{endo-LTD}}$, and was adjusted to make the resultant LTD expression similar to experimentally observed LTD (Supplementary Fig. 9a). When different values of $\tau$ were used to test the different speed of LTD expression, the integration of $k_{\mathrm{endo-LTD}}$ was kept constant (Supplementary Fig. 9b). Thus, AMPAR trafficking in this system was formulated by the following equations:

$$\frac{\mathrm{d}[M]}{\mathrm{d}t} = -k_{\mathrm{out}}[M] + k_{\mathrm{in}}[E] \tag{4}$$

$$\frac{\mathrm{d}[E]}{\mathrm{d}t} = -(k_{\mathrm{endo}} + k_{\mathrm{endo-LTD}} + k_{\mathrm{in}})[E] + k_{\mathrm{out}}[M] + k_{\mathrm{exo}}[I] \tag{5}$$

$$\frac{\mathrm{d}[I]}{\mathrm{d}t} = -k_{\mathrm{exo}}[I] + (k_{\mathrm{endo}} + k_{\mathrm{endo-LTD}})[E] - S \cdot A([E], [I], [V]) \tag{6}$$

$$\frac{\mathrm{d}[V]}{\mathrm{d}t} = S \cdot A([E], [I], [V]) \tag{7}$$

The third factor added in the model is the process of accumulation in the vacuolar portion and LE sorting of AMPARs. The accumulation is modeled as the function $S \cdot A([E],[I],[V])$, where $S$ is a scaling factor. It has been shown in cell biological studies that the accumulation of Rab5 on endocytic vesicles and EE precedes its sudden conversion to Rab7 at the time of LE sorting, and that Rab5-Rab7 conversion is required for sorting from EE to LE[29]. This suggests that the accumulation and removal of Rab5 represent the accumulation and LE sorting of endocytic vesicles, including endocytosed AMPARs. Therefore, we mimicked the dynamics of Rab5 to describe the process of accumulation and LE sorting of AMPARs in the model, and constructed $A([E],[I],[V])$ as the following formula:

$$A([E], [I], [V]) = k_{\mathrm{acc}}[I] \cdot \frac{(k_{\mathrm{endo}} + k_{\mathrm{endo-LTD}})[E]}{1 + e^{-\frac{(k_{\mathrm{endo}} + k_{\mathrm{endo-LTD}})[E] - a}{b}}} \cdot [V](V_{\mathrm{max}} - [V]) \tag{8}$$

where $k_{\mathrm{acc}}[I]$ represents the linear property of the trafficking within the intracellular pool. The second term was included to describe the property of AMPAR accumulation in the vacuolar portion that depends greatly on the enhanced endocytosis during LTD expression, but little on the constitutive endocytosis. The final term, which reflects the sigmoidal accumulation of AMPARs, was constructed based on the autocatalytic property of Rab5 accumulation[40], and $V_{\mathrm{max}}$ presented the maximum capacity to accept AMPARs in the vacuolar portions.

The switch of LE sorting was implemented by resetting the value $[V]$ at a threshold ($h$) to the initial value (0.0001), which represents little accumulation at the basal level. This simple implementation allows us to essentially mimic the steep disappearance of Rab5 from the vacuolar portion that was previously described in a model of Rab5-Rab7 conversion based on the electric cut-out switch[40].

To test the effects of the varied speed of LTD expression or threshold of LE sorting, we used different values of the peak time of $k_{\mathrm{endo-LTD}}$ ($\tau$) and of the threshold of LE sorting ($h$). The values of $\tau$ used were within a range in which the resultant LTD expression was reasonable in view of the experimental results. The values of $h$ used were between the $V_{\mathrm{max}}$ and minimum values, the latter of which were determined as $[V]$ when the slope in the curve of $[V]$ became <0.015. The total number of examples tested with different combinations of $\tau$ and $h$ were 12,292. For the analyses of the recovery and LTD groups upon the different timing of inhibition of LE sorting, we picked a set of 50 examples from all the examples by simple random sampling techniques using MATLAB software. For the comparison of modeling results with experimental results, the speed of expression ($\alpha$) was calculated in a similar way to the experimental analysis: time course of normalized AMPAR numbers for the first 8 min was fitted to a linear function ($f(t) = -\alpha t + \beta$), and the magnitude of slopes ($\alpha$) were used as the speed of expression.

**Data availability.** All data supporting the findings of this study are available within the article and its Supplementary Information or from the corresponding authors upon reasonable request.

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

## Acknowledgements

We thank Dr. George J. Augustine and Dr. Fidel Santamaria for their valuable comments regarding the manuscript, and Ms. Yoonhee Kim for technical assistance. This work was supported by the World Class Institute (WCI) Program of the National Research Foundation of Korea (NRF) funded by the Ministry of Education, Science and Technology of Korea (MEST, NRF Grant Number: WCI 2009-003), the Korea Institute of Science and Technology Institutional Program (Project No. 2E26190), and the NRF grant funded by MEST (No. 2016008165).

## Author contributions

All authors designed the study, performed experiments, analyzed data, and wrote the manuscript. T.K. constructed the computational model.

## Additional information

**Competing interests:** The authors declare no competing financial interests.

