## [Peer Review File · Nature Communications]

Reviewers' comments:

Reviewer #1 (Remarks to the Author):

In this manuscript, Kim et al. study the alterations of AMPA receptor trafficking during long term depression (LTD) in parallel fiber synapses in Purkinje cells, an important and well-studied model of synaptic plasticity. It has long been recognized that postsynaptic AMPA receptors are constantly internalized and recycled, revealing the existence of a pool of intracellular receptors that are in equilibrium with receptors at the surface of Purkinje cells, including synaptic receptors. In this manuscript, the authors formulate and test, with electrophysiology, modelling and novel photoactivatable tools, the hypothesis that the pool of intracellular receptors is reduced during LTD because part of intracellular receptors are removed from the intracellular pool with a mechanism which depends on Rab7. This mechanism was already reported for hippocampal LTD (Fernandez-Monreal et al. 2012, cited in the manuscript) and it is now shown to have a similar role in cerebellar LTD. The novelty of this work resides in the use of a photoactivatable dominant negative form of Rab7 called IE-LOV-Rab7TN, hence a measure of the dynamics of Rab7 function after LTD induction and the integration of these kinetic data into a model of receptor trafficking. These results are potentially of high significance for the molecular and cellular mechanisms of LTD. However, the authors need to provide a number of experimental clarifications to make this manuscript suitable for publication.

Main points:

1. The first experimental evidence for the reduction of an intracellular pool of receptors is that the infusion of TeTx reduces EPSC amplitude less after than before chemically induced LTD. First the authors need to provide evidence that in their conditions LTD occurs. The amount of LTD observed will also be important to calibrate the estimates of intracellular pool size with TeTx, and to perform important control experiments thereafter. Second they propose that the kinetics of this reduction is similar before and after LTD, and that this is consistent with a lack of change in exo-endocytosis rates. However, TeTx needs to diffuse within the cell and cleave synaptobrevin to block exocytosis. The rate of EPSC decrease may thus be dominated by these parameters and will not reflect the rate of internalization. The authors need to provide another argument to rule out the possibility that endocytosis is affected or evoke the possibility that these rates are also affected.
2. The complete lack of depression after induction with Rab7TN is at odds with the model in which sorting of receptors to LEs is secondary to receptor endocytosis. If kendo is really accelerated for minutes (see revised model), there should be a transient decrease in EPSCs. The authors need to consider indirect effects of Rab7TN, for example on earlier steps such as receptor internalization.
3. The Rab7 constructs need to be better characterized. The in vitro effector binding assays to RILP are made on the Rab7, not the mutated Rab7TN. Therefore, even if the LOV domain fusions are interfering with RILP binding, it is only indirectly related with the putative function of LOV-Rab7TN constructs. Moreover, the sorting and degradation of receptors need to be assessed directly, at least in transfected cell lines, or perhaps better in neuronal cells, but this might be technically challenging. They also need to check the effect of illumination, not only the "active" and "inactive" mimics CA-LOV-RAB7TN and IE-LOV-RAB7TN. It is important to calibrate the effects of illumination of the LE redirection of receptors.
4. Some of the parameters for the modelling are not clear. How is speed of expression calculated? It is calculated by "fitting the time course of normalized PF-EPSCs to a linear function". What is the unit? Is the extent of LTD related to this 'speed'? The involvement of kendoLTD in the model is not clear. The way it affects the measure, and its kinetics, should be better explained, perhaps with a graph, showing basal kendo for comparison. In general, the figure legends do not describe with enough precision the parameters and how they fit the experimental data.

Minor points

1. In many cases the graphs do not have enough ticks to correctly measure the parameters. For example in Figure 1B-D.

2. In figure 6, why is the effect of BLPA between 13-18 min (in figure 5) not represented? It would allow direct comparison of all the intervals.
3. Some of the many acronyms used in the manuscript are unnecessary long (BLPA, PF&dV) making the reading difficult at times

Reviewer #2 (Remarks to the Author):

In this manuscript, Kim et al developed a novel optogenetic tool, a photosensitive dominant-negative Rab7, to investigate the role of late endosomal trafficking of AMPAR in the maintenance of cerebellar LTD. Combining electrophysiological recording and computational modeling, they report that transient late endosome sorting is required for proper LTD maintenance at PF-PC synapses. This genetic-coded, photosensitive Rab7 potentially can be used in various cell types/systems and thus can be of interest to many others in other fields. Another contribution of this study is the insight on the temporal requirement of late endosomal trafficking in cerebellar LTD maintenance. However, this study relies solely on the manipulation of Rab7 and uses AMPAR currents as the only readout. The key question that how Rab7 affects AMPAR trafficking and LTD is not well characterized. There are also substantial concerns about the AMPAR recycling experiments and the interpretation of the data. Without better-defined mechanistic studies, this paper reads stronger as a method paper but falls short on the biological conclusions. Additional experiments are needed to fill the gap in understanding the action of Rab7 and its link it to AMPAR trafficking during LTD maintenance.

1. In the TeTx experiment, the author interpreted T1/2 (~7.5 min) as the speed of AMPAR internalization. However, delivering of TeTx via a patch pipette into a large cell like a Purkinje cell takes at least a few minutes which is expect to affect T1/2 quite significantly. Since the author also infused Alexa Flour along with TeTx, plotting fluorescent intensity of Alexa Flour of the dendrite/spine area where the stimulation electrode is may help to estimate the contribution of toxin infusion vs. AMPAR internalization to T1/2.
2. AMPAR exocytosis involves both the recycling and secretory pathways. Does TeTx block only the recycling-dependent exocytosis but not the other? If yes, please specificity the reference or provide experimental evidence. If not, what is the contribution of secretory exocytosis to MR and how does it affect the modeling?
3. The involvement of AMPAR recycling in LTD has been demonstrated in hippocampus (Fernandez-Monreal et al., 2012) in which dominant Rab11 facilitates and doubles NMDAR-dependent LTD. While Rab11 activity appears to be stable during LTD in the previous study, the authors claims that AMPAR recycling is decreased during mGluR-LTD maintenance. What are the authors' thoughts on this discrepancy?
4. The recycling part of the story seems to be disconnected to the rest of the study. It used chemically induced LTD (K+ & Glutamate) at least 30 mins after the induction but the Rab7 manipulations showed a transient effect between 8-23 after the electrical (FP&dV) induction. These two types of induction protocols can be very different in AMPAR trafficking kinetics and signaling, and are thus not comparable. It weakens the authors' claim that the transient activation of late endosome sorting, proceeding the reduction of AMPAR number in the recycling pool, functions as a switch to keep synaptic depression during LTD maintenance.
5. The PS-Rab7TN was only characterized using lysotracker and should be characterized using other sorting assays to measure sorting into lysosomes.
6. The authors link the major effect of PS-Rab7TN to lysosome intensity in HeLa cells. Does the lysosome intensity decrease in PCs? A comparison of lysotracker intensity before and after BLPA on PCs will be informative. Several lines of evidence also show that Rab7 can regulate retromers but the potential interference of the retrograde trafficking in Hela or PCs is not examined or discussed.
7. One major finding of this study is the transient disruption of LTD with PS-Rab7-DN. Interestingly,

after the 5-min BLPA activation of Rab7-DN (anytime between 8-23 min especially in figure6), AMPAR currents appears to be stabilized (neither return to the baseline nor decline to the degree of slices without BLPA), suggesting there is an alteration of non-synaptic AMPAR accumulation or AMPAR internalization. Additionally, the influence of Rab7-regulated late endosomal trafficking presumably is not restricted to AMPARs. Did the authors consider the involvement of lysosomal degradation of other proteins during LTD maintenance (for example scaffold proteins of AMPAR that can impact AMPAR trafficking)? In general, additional experiments are needed to better define the action of PS-Rab7 on AMPAR trafficking and LTD.

8. The authors chose LOV-Rab7-546-2 for the LTD study after testing various constructs with different junction sequences. The binding properties LOV-Rab7-546-2 to RBD-RILP should be included.

9. The immunohistochemistry of endogenous Rab7 looks very spine enriched in PCs. What are the expression patterns for CA-Rab7-TN, IE-Rab7-TN and PS-Rab7-TN in PCs? Does BLPA induce subcellular distribution changes in PS-Rab7-TN?

Reviewer #3 (Remarks to the Author):

Kim et al. report on the mechanisms underlying the maintenance of parallel fiber-Purkinje cell LTD. They propose and test a mechanism of LTD maintenance involving a reduction of the number of AMPA receptors available for trafficking into the postsynaptic membrane. They develop a photosensitive inhibitor of endosome sorting and also provide a simple model of AMPA receptor trafficking that supports and explains their findings. Their work suggests a mechanism for an LTD threshold and may have implications for the memory storage capacities of Purkinje cells. Overall the work is novel and clearly presented. The issue of how memories are stored in the cerebellum is an important one. I have just a few relatively minor comments.

1) I am unclear why Rab7TN, a blocker of vesicle sorting, completely blocks expression and maintenance of LTD (figure 3E). Isn't it expected that only the maintenance should be blocked? I thought expression was due to PKC and map kinase signaling. This should be discussed.

2) A point relating to framing of results. At bottom of page 14 authors transition to more in depth analysis of the data involving the BLPA. The motivation for this might be more clear if the authors point out the fact that the blockade of LTD appears to be incomplete in all cases and they wanted to understand why. The more in depth analysis shows that this is because of two populations of responses, ie cells in which LTD is completely blocked or completely unaffected.

3) Some brief discussion of functional implications might be useful, eg multi-state synapses have important computational advantages for memory storage capacity as shown by Fusi and others. Mechanisms described here may relate to this.

4) Under natural conditions LTD is likely induced by paired parallel fiber and climbing fiber stimulation. Did the authors try any of their manipulations on climbing fiber induced-LTD? If not, why and do they think the results would be the same?

REVIEWERS' COMMENTS:

Reviewer #1 (Remarks to the Author):

1. I agree with the explanations provided concerning the diffusion of TeNT into cells and the comparison of rates of decrease to determine k_{endo} and N_{tot} . However, I think adding the data in additional figure 1 of the rebuttal letter to a supplementary figure in the manuscript would help the reader to know exactly how the measurements were obtained.
2. I agree with the author's response on this point and with the added text in the discussion.
3. I appreciate the effort of the authors in further validating the PS-Rab7TN. This is now demonstrated much more convincingly that it is able to affect after photoactivation the transfer of cargo to lysosomes (sup Figure 5). Even if this assay was not performed on neurons it is now convincing that the effect of PS-Rab7TN and illumination is to perturb sorting of cargo to late endosomes or lysosomes.
4. The modelling and analysis is now much clearer. Supplementary Figure 9 helps to understand the interplay between the increase in endocytosis and the transition from early to late endosomes. Perhaps in the discussion the authors could refer to two recent publications showing that an LTD protocol in hippocampal cultured neurons transiently increases AMPAR endocytosis, similar to the prediction of the model (Rosendale et al. Cell Reports 2017; Fujii et al. Genes Cells 2017). Finally, all the minor points were corrected appropriately in the new version.

Reviewer #3 (Remarks to the Author):

The authors have addressed my concerns. I find the revised ms improved and suitable for publication.

Responses to Reviewers' Comments

We thank the reviewers for their constructive suggestions for improving our paper. We have addressed to and incorporated the suggestions. Our detailed responses are listed below, and additional figures are attached at the end of responses for better understanding of our explanation, although these figures are not included in the revised manuscript. The addition or modification we have made in the revised manuscript is described in *italic*.

Reviewer #1 (Remarks to the Author):

In this manuscript, Kim et al. study the alterations of AMPA receptor trafficking during long term depression (LTD) in parallel fiber synapses in Purkinje cells, an important and well-studied model of synaptic plasticity. It has long been recognized that postsynaptic AMPA receptors are constantly internalized and recycled, revealing the existence of a pool of intracellular receptors that are in equilibrium with receptors at the surface of Purkinje cells, including synaptic receptors. In this manuscript, the authors formulate and test, with electrophysiology, modelling and novel photoactivatable tools, the hypothesis that the pool of intracellular receptors is reduced during LTD because part of intracellular receptors are removed from the intracellular pool with a mechanism which depends on Rab7. This mechanism was already reported for hippocampal LTD (Fernandez-Monreal et al. 2012, cited in the manuscript) and it is now shown to have a similar role in cerebellar LTD. The novelty of this work resides in the use of a photoactivatable dominant negative form of Rab7 called IE-LOV-Rab7TN, hence a measure of the dynamics of Rab7 function after LTD induction and the integration of these kinetic data into a model of receptor trafficking. These results are potentially of high significance for the molecular and cellular mechanisms of LTD. However, the authors need to provide a number of experimental clarifications to make this manuscript suitable for publication.

Response:

We thank the reviewer for finding the potential significance in our study as well as important suggestions for improving our manuscript. We performed control experiments and computational analyses, and incorporated them into our manuscript, in accordance with the reviewer's suggestions.

Main points:

1. The first experimental evidence for the reduction of an intracellular pool of receptors is that the infusion of TeTx reduces EPSC amplitude less after than before chemically induced LTD. First the authors need to provide evidence that in their conditions LTD occurs. The amount of LTD observed will also be important to calibrate the estimates of intracellular pool size with TeNT, and to perform important control experiments thereafter. Second they propose that the kinetics of this reduction is similar before and after LTD, and that this is consistent with a lack of change in exo-endocytosis rates. However, TeTx needs to diffuse within the cell and cleave synaptobrevin to block exocytosis. The rate of EPSC decrease may thus be dominated by these parameters and will not reflect the rate of internalization. The authors need to provide another argument to rule out the possibility that endocytosis is affected or evoke the possibility that these rates are also affected.

Response:

In our previous studies, we have shown that treatment of slices with 50 mM K⁺ and 10 μM glutamate (K-glu) for 5 min indeed induces LTD by recording evoked EPSCs and changing extracellular solutions to K-glu for 5 min (Tanaka and Augustine, 2008), or by recording miniature EPSCs at 30-60 min after treating slices with K-glu for 5 min (Yamamoto et al., 2012).

In the former study, input resistance and membrane current were substantially altered immediately after K-glu treatment, presumably due to the K-glu treatment causing a temporary activation of ion channels. Nevertheless, they were relatively stable at around 40 min, and the amount of LTD was 32.3% at 40-60 min after the K-glu treatment. This amount of LTD (~30%) was considered to choose a specific value of reduced N_{tot} or increasing k_{endo} to evaluate whether either change of parameters could reproduce experimentally obtained results of TeTx-mediated PF-EPSC reduction, as shown in Figure 2B-2E.

Although we originally referred our previous studies, *we modified the description to clearly state that LTD is indeed induced by K-glu treatment, in the revised manuscript (p.6, lines 14-16). We also mentioned that the amount of PF-EPSC reduction by the specific value of reduced N_{tot} ("a" shown in Figure 2B) is equivalent to the K-glu-induced LTD (p.8, line 6).*

Regarding the second point, we first have to clarify the time when PF-EPSC reduction was presented in Figure 1B of our manuscript. The PF-EPSC amplitudes usually started to decrease around 6-8 min after whole cell configuration with internal solution including TeTx, similar to a previous report (Tatsukawa et al., 2006). We assume that TeTx diffuses into stimulated dendrites during this period, and its concentration reaches a level that causes the EPSC reduction. The Figure 1B in our manuscript shows the time course of PF-EPSC reduction from right before the start of reduction. *In order to clarify how we observed TeTx-dependent reduction in PF-EPSC, we added descriptions in the Methods (p.28, lines 13-16) and in the Figure legend of Figure 1B.*

Once the PF-EPSC started to decrease, we assume that the kinetics of PF-EPSC decrease rely largely on the AMPAR internalization, because of the following supporting evidences:

- (i) A previous study used two concentrations of TeTx (150 and 300 nM) in Purkinje cells, and showed that 300 nM TeTx did not accelerate the speed of TeTx-mediated EPSC reduction (Tatsukawa et al., 2006). This suggests that TeTx blocks exocytosis nearly in an all-or-none manner, rather than linearly in a concentration-dependent manner, once the TeTx concentration reaches a threshold that causes the EPSC reduction.
- (ii) Antibody-feeding experiments in cultured neurons (Ehlers, 2000; Waung et al., 2008) estimated the rate of AMPAR endocytosis at the basal state. The rate estimated in these studies was indicated as time constant ($1/k$) of about 10 min, which can be converted to $T_{1/2}$ ($(\ln 2)/k$) of 7 min. The speed of TeTx-mediated reduction in PF-EPSC shown in our study ($T_{1/2} = 7.5$ min) is consistent with this value, suggesting that the speed of PF-EPSC reduction observed in our study is determined largely by the AMPAR endocytosis rate, and less by the rate of TeTx diffusion.

In the revised manuscript, we explained why we assumed that $T_{1/2}$ of PF-EPSC reduction relies on the rate of AMPAR internalization, by referring these papers (p.8, 2nd paragraph).

However, it is still possible that the rate of TeTx-dependent blockade of exocytosis would slightly affect the PF-EPSC reduction observed in our study. We therefore included the gradual exocytosis blockade with a half decay period of 5 min in our model, and tested whether our proposal is still reasonable, even if TeTx gradually blocks exocytosis. In order to reproduce the experimental results of TeTx-dependent reduction in PF-EPSC by the modified model, we needed to increase the k_{endo} value, which resulted in the estimation of

the basal AMPAR endocytosis rate as $T_{1/2}$ of 2.8 min. This is 2.5 times faster than reported rate (Ehlers, 2000; Waung et al., 2008), so that the gradual exocytosis blockade with a half decay period of 5 min would be less likely. Even in such extreme situations, the modified model still leads to the same conclusion that the reduction of N_{tot} , but not an increase in k_{endo} , could reproduce experimentally obtained results. *In the revised manuscript, we included the modeling results with the consideration of gradual exocytosis blockade (p.8, 4th line from the bottom – p.9, 1st paragraph, Supplemental figure 1), in order to confirm our conclusion that N_{tot} , but not k_{endo} , is altered during LTD maintenance, even if TeTx gradually blocks exocytosis.*

2. The complete lack of depression after induction with Rab7TN is at odds with the model in which sorting of receptors to LEs is secondary to receptor endocytosis. If kendo is really accelerated for minutes (see revised model), there should be a transient decrease in EPSCs. The authors need to consider indirect effects of Rab7TN, for example on earlier steps such as receptor internalization.

Response:

As this reviewer expected, we agree that the long-term Rab7TN expression may cause changes in other parts of the endosomal pathway, due to continuous interruption of sorting to LEs or lysosomes. Indeed, it was reported that the recycling of nutrient transporter was increased in culture cells expressing Rab7TN (Edinger et al., 2003). Further, our results of unchanged LTD after the photoactivation of PS-Rab7TN at 0-5 min suggest that the complete lack of transient depression in Purkinje cells expressing Rab7TN is due to an indirect effect of Rab7TN expression. *In the revised manuscript, we included the possibility of the indirect effect and the supporting data of photoactivation at 0-5 min in the Discussion (p.21, first 4 lines in 2nd paragraph & p.22, first 3 lines).*

3. The Rab7 constructs need to be better characterized. The in vitro effector binding assays to RILP are made on the Rab7, not the mutated Rab7TN. Therefore, even if the LOV domain fusions are interfering with RILP binding, it is only indirectly related with the putative function of LOV-Rab7TN constructs. Moreover, the sorting and degradation of receptors need to be assessed directly, at least in transfected cell lines, or perhaps better in neuronal cells, but this might be technically challenging. They also need to check the effect of illumination, not only the “active” and “inactive” mimics CA-LOV-RAB7TN and IE-LOV-RAB7TN. It is important to calibrate the effects of illumination of the LE redirection of receptors.

Response:

It has been consistently shown that Rab7 is necessary for endosomal maturation from early endosomes to LE and lysosomes. Rab7TN cannot be active due to lack of GTP binding, so that dominantly expressed Rab7TN prevents the pathway in the endosomal maturation or molecular trafficking where Rab7 activity is required (Feng et al., 1995; Bucci et al., 2000; Vonderheit and Helenius, 2005). However, the exact mechanism of Rab7TN to prevent the pathway and its interacting molecules during the process still remain unidentified. Therefore, biological assays are required to characterize LOV-Rab7TN, and LysoTracker staining was originally performed. To address this reviewer’s suggestion, we performed a new analysis, low-density lipoprotein (LDL) uptake assay in HeLa cells, and tested the effects of photoactivation of PS-Rab7TN.

It has been shown that LDL is transported into lysosomes in HeLa cells, when LDL is applied to the culture medium of HeLa cells that has been kept in LDL deficient medium for

18-24 hours (Bucci et al., 2000). We used DyLight550-labeled LDL (LDL-DL), and stained cells with an antibody of Lamp2, a lysosomal marker. We then found that colocalization of LDL-DL with Lamp2 was reduced in cells expressing Rab7TN compared with cells expressing GFP alone. The colocalization in cells expressing PS-Rab7TN was comparable to that in cells expressing GFP, yet applying photoactivation to cells expressing PS-Rab7TN caused reduction of colocalization, similar to the level seen in cells expressing Rab7TN.

As the reviewer expected, the analysis was challenging. We needed to establish a new assay system, which allows us to detect molecular sorting to LEs or lysosomes, while applying photoactivation. For the LTD experiments, we applied local photoactivation ($15 \times 15 \mu\text{m}^2$ ROI) by using scanning laser of confocal microscope. For the LDL uptake assay, because it was not practical to find cells receiving such local photoactivation after the process of immunocytochemistry, LED light source (peak at 470 nm) was used to apply photoactivation to whole areas of cover slips, where cells are seeded. Even though the luminance of photoactivation per area (13 mW in 70 mm^2 sensor, for 60 min) could be 20 times lower than that of the scanning laser, the inhibition of sorting was detected in the assay. Therefore, this calibration supports our assertion that the local photoactivation used for LTD experiments caused the inhibition of sorting toward LEs and lysosomes.

In the revised manuscript, we included the results of LDL uptake assay (Supplemental figure 5, p.13, 2nd paragraph). In addition, we mentioned the comparison of luminance of photoactivation used for LTD experiments and for LDL uptake assay in the Results as well as the Methods (p. 15, lines 4-6 & p. 31, lines 7-10).

4. Some of the parameters for the modelling are not clear. How is speed of expression calculated? It is calculated by “fitting the time course of normalized PF-EPSCs to a linear function”. What is the unit? Is the extent of LTD related to this ‘speed’? The involvement of kendoLTD in the model is not clear. The way it affects the measure, and its kinetics, should be better explained, perhaps with a graph, showing basal kendo for comparison. In general, the figure legends do not describe with enough precision the parameters and how they fit the experimental data.

Response:

When the speed of LTD expression was calculated from the experimental results, we used data points during the first 8 min after PF& Δ V, in order not to include the effects of photoactivation. Because of small numbers of the data points, we used a linear function to fit the time course of normalized PF-EPSCs, and then had the magnitude of slope (α) of linear function ($f(t) = -\alpha t + \beta$) as the speed of expression. *In the revised manuscript, we added the unit of the speed of LTD expression, % reduction of PF-EPSCs/min, in the Figure legends of Figure 8, and described the extent of LTD used for fitting and calculation of the speed of LTD expression in the Results as well as the Methods. (p.18, 7th line from the bottom, & p.28, 4th line from the bottom).*

In the original manuscript, we actually showed the speed of expression calculated by fitting to the data points before the photoactivation (0-8, 0-13, or 0-18 min). Although this calculation might be reasonable to compare the speed of individual results among the same photoactivation times (Figure 8A), it is not appropriate to compare the speed between different photoactivation times (Figure 8B). We thus reanalyzed by fitting to the data points during the first 8 min after PF& Δ V for all photoactivation times, and found that the results were still the same: LTD with the similar speed could fall into either the recovery or LTD group, yet the speed of LTD expression tended to correlate with the timing of photoactivation.

In order to obtain modeling results comparable to the experimental results, we also used a linear function ($f(t) = -\alpha t + \beta$) to fit the normalized AMPAR numbers at synapses during the first 8 min, and estimated the speed of expression (α). We added the unit (% reduction of PF-EPSCs/min) in the Figure legend of Figure 8, and described how to calculate the speed of expression from the modeling results in the Methods of the revised manuscript (p.38, last 3 lines – p. 39, first 2 lines).

To clarify the involvement of $k_{\text{endo-LTD}}$ described by a piecewise-defined concave function, we have added Supplemental figure 9A and 9B. The Supplemental figure 9A shows changes in total endocytosis rate ($k_{\text{endo}} + k_{\text{endo-LTD}}$) and resultant normalized AMPAR numbers at synapses with or without $k_{\text{endo-LTD}}$. We mentioned the Supplemental figure 9A, when we first described $k_{\text{endo-LTD}}$ in the Result section (p.19, line 17). The Supplemental figure 9B shows the different values of τ (peak time of $k_{\text{endo-LTD}}$) resulting in the different speed of LTD expression, and was mentioned in the first description of τ in the Result section (p.20, first line). Also, we added explanation in the Methods (p36, last 3 lines – p.37, first line) with the indication of Supplemental figure 9A and 9B.

For better description of parameters, we added specific parameters used in Figure 8D and new Supplemental figure 9, in their Figure legends. We also included Supplemental table 1, which shows the original parameters of the model, in the Figure legend of Figure 2.

Minor points

1. In many cases the graphs do not have enough ticks to correctly measure the parameters. For example in Figure 1B-D.

Response:

We added ticks in Figure 1B-1D as well as others (e.g. Figure 2C-2E, Figure 4C, Figure 6A-6F, Figure 7C-7E, Figure 8F).

2. In figure 6, why is the effect of BLPA between 13-18 min (in figure 5) not represented? It would allow direct comparison of all the intervals.

Response:

As suggested, we added the results of BLPA at 13-18 min in Figure 6E and 6F.

3. Some of the many acronyms used in the manuscript are unnecessary long (BLPA, PF&dV) making the reading difficult at times

Response:

We changed BLPA to PA and PF& Δ V to PF Δ V.

Reviewer #2 (Remarks to the Author):

In this manuscript, Kim et al developed a novel optogenetic tool, a photosensitive dominant-negative Rab7, to investigate the role of late endosomal trafficking of AMPAR in the maintenance of cerebellar LTD. Combining electrophysiological recording and computational modeling, they report that transient late endosome sorting is required for proper LTD maintenance at PF-PC synapses. This genetic-coded, photosensitive Rab7 potentially can be used in various cell types/systems and thus can be of interest to many others in other fields. Another contribution of this study is the insight on the temporal requirement of late endosomal trafficking in cerebellar LTD maintenance. However, this study relies solely on the manipulation of Rab7 and uses AMPAR currents as the only readout. The key question that how Rab7 affects AMPAR trafficking and LTD is not well characterized. There are also substantial concerns about the AMPAR recycling experiments and the interpretation of the data. Without better-defined mechanistic studies, this paper reads stronger as a method paper but falls short on the biological conclusions. Additional experiments are needed to fill the gap in understanding the action of Rab7 and its link it to AMPAR trafficking during LTD maintenance.

Response:

We thank the reviewer for efforts to evaluate our paper and for many constructive suggestions to improve our manuscript. However, there seem to be some misleading points to be clarified. We would like to specifically emphasize that we have done detailed analyses of experimental results as well as computational analyses to define when and how LE sorting works for LTD. We believe that these analyses can be considered as a well-defined mechanistic study, and the results are biologically important, since our study determined the properties of LE sorting, a ubiquitous biological event, that accomplishes the transition into the maintenance of cerebellar LTD. Further, this study was only possible by using our newly developed photosensitive tool, PS-Rab7TN, which allows us to temporally control the LE sorting during electrophysiological LTD recording in cerebellar slices.

While we agree with this reviewer that it is also interesting to investigate molecular mechanisms as to how Rab7 affects AMPAR trafficking and how the properties of LE sorting during LTD can be explained, we feel that our paper already is a full, complete analysis of timing and properties of LE sorting required for LTD. Furthermore, we believe that the investigation of molecular mechanisms would require a new tool with a temporal control as well as a substantial amount of experiments that would be justified as a separate project by itself.

We have improved our paper by addressing all of the comments made by this reviewer and by clarifying the points that potentially caused the confusion, as described below.

1. In the TeTx experiment, the author interpreted T1/2 (~7.5 min) as the speed of AMPAR internalization. However, delivering of TeTx via a patch pipette into a large cell like a Purkinje cell takes at least a few minutes which is expect to affect T1/2 quite significantly. Since the author also infused Alexa Flour along with TeTx, plotting fluorescent intensity of Alexa Flour of the dendrite/spine area where the stimulation electrode is may help to estimate the contribution of toxin infusion vs. AMPAR internalization to T1/2.

Response:

The rate of AMPAR endocytosis was previously measured in studies using antibody-feeding experiments in cultured neurons (Ehlers, 2000; Waung et al., 2008). These studies demonstrated that the rate was indicated as time constant ($1/k$) of about 10 min, which can be converted to $T_{1/2}$ ($(\ln 2)/k$) of about 7 min, similar to $T_{1/2}$ of TeTx-induced PF-EPSC reduction in our study (7.5 min). *In the revised manuscript, we referred these papers, to support our estimation of AMPAR endocytosis rate from TeTx-dependent reduction in PF-EPSC (p.8, lines 18-20).*

Nevertheless, it is still important to consider the diffusion of TeTx, as the reviewer has pointed out. We appreciate the suggestion of using the Alexa dye imaging to estimate the diffusion of TeTx. The diffusion of Alexa dye and probably TeTx was actually very slow (time to reach half-maximum, 15 min, Additional figure 1). However, we believe that the rate of TeTx-dependent blockade of exocytosis dissociates from the rate of diffusion, because of the following supporting evidences:

- (i) As described in the response to second point of comment 1 from reviewer #1, a previous study used two concentrations of TeTx (150 and 300 nM) in Purkinje cells, and showed that 300 nM TeTx did not accelerate the speed of TeTx-mediated EPSC reduction (Tatsukawa et al., 2006). This suggests that TeTx blocks exocytosis nearly in an all-or-none manner.
- (ii) As mentioned above, $T_{1/2}$ of TeTx-mediated EPSC reduction in our study is equivalent to the $T_{1/2}$ of AMPAR endocytosis rate measured by antibody-feeding experiments (Ehlers, 2000; Waung et al., 2008), suggesting that $T_{1/2}$ of TeTx-mediated EPSC reduction largely relies on the rate of AMPAR endocytosis.
- (iii) We also tried to include the slow kinetics (half decay period of 15 min) of exocytosis blockade into our model by considering the speed of diffusion of Alexa dye. We then found that the experimental results of TeTx-dependent reduction in PF-EPSC could be reproduced by the model with an extremely large k_{endo} value (200 min^{-1} instead of 0.13 min^{-1} , Additional figure 2). The large k_{endo} value is viewed as the basal AMPAR endocytosis rate of $T_{1/2} = 0.7 \text{ min}$ (Additional figure 3). This rate is 10 times faster than the rate measured in antibody-feeding experiments (Ehlers, 2000; Waung et al., 2008), so that such slow exocytosis blockade is not realistic.

These evidences suggest that the kinetics of TeTx-dependent blockade of exocytosis is likely to be faster than the diffusion of intracellular solutions, and that TeTx would block exocytosis nearly in an all-or-none manner.

We thus modified our model by including the assumption of faster kinetics of exocytosis blockade (half decay period of 5 min). This model estimated the rate of AMPAR endocytosis as $T_{1/2} = 2.8 \text{ min}$, which is 2.5 times faster than the reported rate, yet led to the same conclusion that N_{tot} , but not k_{endo} , is altered during LTD maintenance. *We added the new analysis in the revised manuscript (p.8, 4th line from the bottom – p.9, 1st paragraph, Supplemental figure 1).*

As we also mentioned in the response to comment 1 from reviewer #1, we did not accurately describe in the original manuscript about the period when TeTx diffuses into Purkinje cells through patch pipette. The PF-EPSC amplitudes started to decrease around 6-8 min after whole cell configuration, suggesting that TeTx diffuses into stimulated dendrites during this period. The Figure 1B in our manuscript shows the time course of PF-EPSC reduction from right before the start of reduction. *In the revised manuscript, we described the time when TeTx-dependent reduction in PF-EPSC was observed in the Methods (p.28, lines 13-16) and in the Figure legend of Figure 1B.*

2. AMPAR exocytosis involves both the recycling and secretory pathways. Does

TeTx block only the recycling-dependent exocytosis but not the other? If yes, please specify the reference or provide experimental evidence. If not, what is the contribution of secretory exocytosis to MR and how does it affect the modeling?

Response:

Considering that synaptobrevin is likely to be involved also in the transport of molecules to plasma membrane through secretory pathway, TeTx may also block AMPAR exocytosis through the secretory pathway, although it is not clearly reported.

It has been reported that the half-life of AMPAR subunit proteins is several tens of hours to days (Archibald et al., 1998; Kjølner and Diemer, 2000; Cohen et al., 2013), so that the kinetics of secretory pathway seems to be very slow in comparison to the time scale we handled (~ an hour). In fact, addition of the secretory pathway ($T_{1/2} = 47$ hours, Cohen et al., 2013) in our model without modifying other parameters had a limited impact on the EPSC reduction regardless of whether the pathway was blocked with TeTx or not (Additional figure 4). Therefore, we did not further consider the secretory pathway. *We have added a description of this model result (data not shown) in the Methods of revised manuscript, to explain why we did not include the slow kinetics of secretory pathway in the model (p.36, 2nd paragraph).*

3. The involvement of AMPAR recycling in LTD has been demonstrated in hippocampus (Fernandez-Monreal et al., 2012) in which dominant Rab11 facilitates and doubles NMDAR-dependent LTD. While Rab11 activity appears to be stable during LTD in the previous study, the authors claims that AMPAR recycling is decreased during mGluR-LTD maintenance. What are the authors' thoughts on this discrepancy?

Response:

We think that there is no discrepancy between our results and the previous study of hippocampal LTD (Fernandez-Monreal et al., 2012). We assume that there would be some misunderstanding, so that we would like to explain our results. The experiments using TeTx demonstrated that maximum EPSC reduction ($MR_{PF-EPSC}$) was decreased, while time constant of EPSC reduction ($T_{1/2}$) was not altered during LTD maintenance. Our computational model then showed that the experimental results were reproduced by the reduction of total AMPAR numbers in the recycling loop (N_{tot}), but not by increase in the transition from extrasynaptic to intracellular pools (k_{endo}). These results led us to conclude that numbers of AMPARs residing in the recycling loop was decreased, but the speed of AMPAR recycling (endocytosis) was not altered during the maintenance of cerebellar LTD. Because Rab11 activity accelerates the transport of AMPARs from recycling endosomes to the postsynaptic compartment, the finding of unaltered Rab11 activity during LTD in the previous study suggests that they did not detect changes in the speed of AMPAR recycling. In terms of unaltered speed of AMPAR recycling, our results are consistent with the previous study of hippocampal LTD (Fernandez-Monreal et al., 2012).

To avoid the misunderstanding, we modified "recycling AMPARs are reduced..." to "recycling AMPAR numbers are reduced..." in the abstract (the 3rd line from the bottom of abstract).

4. The recycling part of the story seems to be disconnected to the rest of the study. It used chemically induced LTD (K+& Glutamate) at least 30 mins after the induction but the Rab7 manipulations showed a transient effect between 8-23 after the electrical (FP&dV) induction. These two types of induction protocols can be very

different in AMPAR trafficking kinetics and signaling, and are thus not comparable. It weakens the authors' claim that the transient activation of late endosome sorting, proceeding the reduction of AMPAR number in the recycling pool, functions as a switch to keep synaptic depression during LTD maintenance.

Response:

We have previously shown that chemical LTD stimulation (K-glu) indeed induced LTD (Tanaka and Augustine, 2008; Yamamoto et al., 2012). In addition, the preincubation of slices with K-glu for 5 min occluded PF& Δ V-evoked LTD (Lee et al., 2015), suggesting that two forms of LTD share common signaling pathways.

As the purpose of using K-glu treatment for TeTx experiments is to see AMPAR recycling during LTD maintenance, an important condition is to perform the TeTx experiments during the LTD maintenance. The following evidences support that the experiments were indeed performed during the LTD maintenance:

- (i) We have tested the time course of K-glu-induced LTD (Tanaka and Augustine, 2008). The precise time course of LTD immediately after K-glu treatment was not clarified, because EPSC amplitude was transiently increased presumably due to the K-glu treatment causing temporary changes in passive membrane properties, as is evident in severe alteration of input resistance and membrane current. However, the input resistance and membrane current became relatively stable at around 40 min, and the EPSC amplitude was reduced by 32.3% at 40-60 min after the K-glu treatment, which is equivalent to the amount of LTD triggered by PF& Δ V.
- (ii) We have previously measured a translocation of PKC to the plasma membrane, a hallmark of PKC activation, and have demonstrated that the translocation was peaked at 5 min and reduced at 10 min after K-glu treatment. In addition, immunohistochemical analysis using an antibody of phosphorylated MAPK revealed that MAPK activation in Purkinje cells returned to the basal level at 5-10 min after K-glu treatment (Tanaka and Augustine, 2008). Both PKC and MAPK are important components of the positive feedback loop, which works for LTD expression. Therefore, these results suggest that the LTD expression is terminated at around 20 min after K-glu treatment, similar to the case of LTD triggered by PF& Δ V.

In the revised manuscript, we cited references and mentioned that LTD induced by K-glu treatment is considered to share common signaling pathways with and to proceed similar time course to LTD induced by PF& Δ V (p.6, lines 14-19).

In the original manuscript, we have mentioned that recordings were made following incubation for 30 min in normal ACSF. After the 30 min incubation, it usually took us about 5 min to make the whole cell configuration, and we waited for 6-8 min of TeTx diffusion, so that the PF-EPSC reduction was recorded around 40 min after the K-glu treatment. *In the revised manuscript, we modified the description (p.6, lines 19-20).*

5. The PS-Rab7TN was only characterized using lysotracker and should be characterized using other sorting assays to measure sorting into lysosomes.

Response:

To address this suggestion, we have performed a new analysis, low-density lipoprotein (LDL) uptake assay in HeLa cells, and tested the effects of photoactivation of PS-Rab7TN on the sorting toward lysosomes.

It has been shown that LDL can be uptaken by HeLa cells that are maintained in the lipoprotein-deficient medium for 18-24 h, and is transported to lysosomes (Bucci et al., 2000). We applied LDL-DyLight550 (LDL-DL) to HeLa cells after maintaining them in

lipoprotein-deficient medium, and measured colocalization of LDL-DL with lysosomes stained with a Lamp2 antibody. We found that the colocalization in cells expressing PS-Rab7TN was equivalent to that in cells expressing GFP alone. In contrast, the colocalization was reduced in cells expressing PS-Rab7TN and received photoactivation, similar to the level seen in cells expressing Rab7TN.

Thus, this analysis allows us to characterize PS-Rab7TN further and to confirm that the photoactivation of PS-Rab7TN inhibited the sorting into lysosomes.

In the revised manuscript, we added the results of LDL uptake assay (Supplemental figure 5, p.13, 2nd paragraph).

6. The authors link the major effect of PS-Rab7TN to lysosome intensity in HeLa cells. Does the lysosome intensity decrease in PCs? A comparison of lysotracker intensity before and after BLPA on PCs will be informative. Several lines of evidence also show that Rab7 can regulate retromers but the potential interference of the retrograde trafficking in Hela or PCs is not examined or discussed.

Response:

LysoTracker is a fluorescent acidotropic probe for labeling acidic organelles in live cells, so that LysoTracker is capable of labeling lysosomes, which are acidic organelles. However, because LysoTracker signals are not reversible, and the reduction of LysoTracker staining is presumably a consequence of long-term interruption of endosomal maturation by expression of Rab7TN, it is not possible to see the LysoTracker signal reduction by the photoactivation of PS-Rab7TN. Instead, we have performed LysoTracker analysis in Purkinje cells expressing IE- or CA-LOV-Rab7TN-546-2.

To test LysoTracker signals in Purkinje cells, AAV triggering the expression of GFP, GFP-fused Rab7TN, IE- or CA-LOV-Rab7TN-546-2 was stereotaxically injected into PCP2-Cre mice. Cerebellar slices obtained from these mice were then stained with LysoTracker (Song et al., 2008). Images of fresh cerebellar slices were used to analyze the LysoTracker-positive puncta in GFP-positive Purkinje cell dendrites. This analysis demonstrated that expression of Rab7TN, or IE-, but not CA-, LOV-Rab7TN-546-2 reduced numbers of LysoTracker-positive puncta, in comparison with the puncta in Purkinje cells expressing GFP alone.

In the revised manuscript, we included the results of LysoTracker staining in Purkinje cells of cerebellar slices (Supplemental figure 4C and 4D, p.13, lines 6-9).

As reviewer mentioned, Rab7 seems to also regulate retromer (Jia et al., 2016), which has been shown to be involved in AMPAR trafficking (Tian et al., 2015; Munsie et al., 2016). However, a recent study demonstrated that retromer regulates an increase in synaptic AMPAR numbers, long-term potentiation (Temkin et al., 2017), it is unlikely that LTD was inhibited due to photoactivation of PS-Rab7TN interrupting retromer. *In the revised manuscript, we included the discussion about retromer (p. 23, lines 4-11).*

7. One major finding of this study is the transient disruption of LTD with PS-Rab7-DN. Interestingly, after the 5-min BLPA activation of Rab7-DN (anytime between 8-23 min especially in figure6), AMPAR currents appears to be stabilized (neither return to the baseline nor decline to the degree of slices without BLPA), suggesting there is an alteration of non-synaptic AMPAR accumulation or AMPAR internalization. Additionally, the influence of Rab7-regulated late endosomal trafficking presumably is not restricted to AMPARs. Did the authors consider the involvement of lysosomal degradation of other proteins during LTD maintenance (for example scaffold proteins of AMPAR that can impact AMPAR trafficking)? In general, additional experiments

are needed to better define the action of PS-Rab7 on AMPAR trafficking and LTD.

Response:

The averaged results calculated from all the recordings indeed show that the inhibition of LTD by the transient photoactivation of PS-Rab7TN was only partial (Figure 5E, Figure 6B and 6C). However, we are afraid that our detailed analyses shown in Figure 7 and Supplemental figure 5 (Supplemental figure 8 of the revised manuscript) seem to be overlooked. We have analyzed individual recordings and found that LTD was either almost completely blocked (recovery group) or unaffected by the photoactivation (LTD group). The reason why the inhibition was only partial in the averaged results was because the results were mixture of recovery and LTD groups. Both recovery and LTD groups were observed at all three photoactivation times, 8-13, 13-18, and 18-23 min, but fractions of recovery and LTD groups are not constant among the three photoactivation times, suggesting that LTD in all cells relies on LE sorting, with the timing of sorting varying between cells. Furthermore, by comparing experimental results with computational model, we demonstrated that the varied timing of LE sorting resulted from a combination of the variability in speed of LTD expression and threshold for LE sorting. Therefore, although we did not identify detailed “molecular mechanisms”, we have done detailed analyses of experimental results together with the modeling results, and have shown how and when LE sorting works for LTD.

A part of the reason why our detailed analyses were overlooked may be because the transition from Figure 6 to Figure 7 was not clear. As reviewer #3 suggested, *we added sentences in the revised manuscript, to clarify the motivation of analyses using individual experiments (p.16, first 4 lines of 2nd paragraph).*

We have used PS-Rab7TN as a tool to temporally interrupt the sorting to LE or lysosomes. The use of PS-Rab7TN in our study might give the reviewer impression that Rab7 is specifically regulated for LE sorting during LTD. We think that molecules other than Rab7 probably mediate the specific targeting of AMPARs into LE sorting during LTD, considering that Rab7 is a general regulator for vesicle sorting. Although the molecular mechanism of LE sorting of AMPARs is also interesting to investigate, it is beyond the focus of our study. We therefore have discussed the possible involvement of stargazin and PICK1 in the targeting of AMPARs into LE sorting during LTD. *In the revised manuscript, to clarify the confusion regarding Rab7 action, we further added one sentence before the discussion of stargazin and PICK1 (p.22, lines 9-12), and mentioned that Rab7TN has been used as a tool to block vesicle sorting (p.10, line 5).*

As reviewer expected, there may be other molecules to be sorted into LE together with AMPARs. In the original manuscript, we discussed the possibility that stargazin or PICK1 may mediate the LE sorting of AMPARs, as mentioned above, because they can interact with AMPARs and AP-3, a molecule targeted to LE. In this case, stargazin or PICK1 would be also transported to LEs and lysosomes. *In the revised manuscript, we added the possibility of stargazin or PICK1 being transported to LE together with AMPARs in the Discussion (p.22, 3rd line from the bottom).*

8. The authors chose LOV-Rab7-546-2 for the LTD study after testing various constructs with different junction sequences. The binding properties LOV-Rab7-546-2 to RBD-RILP should be included.

Response:

In the original manuscript, we indeed have shown the binding of LOV-Rab7-546-2 with RBD-RILP in Supplemental figure 2B and 2C (Supplemental figure 3B and 3C of the revised manuscript). The LOV-Rab7-546-2 is a construct including “wild-type” Rab7. For the

LTD experiments, we used LOV-Rab7TN-546-2 as PS-Rab7TN, which includes a dominant negative form of Rab7 (Rab7TN).

The binding assay of wild-type Rab7 with RBD-RILP was performed to test whether LOV- α -fused Rab7 could work as photosensitive Rab7, because a previous study used similar binding assay in order to develop photoactivatable-Rac1. We then found that LOV-Rab7-546-4 resulted in a substantial reduction of binding to RBD-RILP, while binding of IE-LOV-Rab7-564-4 to RBD-RILP recovered to the level of Rab7 binding to RBD-RILP. As shown in Supplemental figure 3B and 3C of revised manuscript, the binding of LOV-Rab7-546-2 to RBD-RILP was much less reduced compared with that of LOV-Rab7-546-4, so that we did not perform further binding assays using IE- or CA-LOV constructs.

We would like to note that Rab7TN does not bind to RBD-RILP even when it is not fused to LOV- α (Additional figure 5, Romero Rosales et al., 2009), so that binding assays cannot be used to test LOV-Rab7TN constructs. Other molecules that Rab7TN interact with to prevent the sorting to LE or lysosomes are not identified. Therefore, we used LysoTracker staining, to test the LOV-Rab7TN constructs.

In the revised manuscript, we modified the description in the Results to clarify that binding assays were performed by using wild-type Rab7, and that another assay was required for testing Rab7TN (p.11, 6th line from the bottom & p.12, first 2 lines of 2nd paragraph).

9. The immunohistochemistry of endogenous Rab7 looks very spine enriched in PCs. What are the expression patterns for CA-Rab7-TN, IE-Rab7-TN and PS-Rab7-TN in PCs? Does BLPA induce subcellular distribution changes in PS-Rab7-TN?

Response:

In the immunohistochemical analysis, Rab7 was found in spine-like protrusions of PCs, but was also found in the soma, proximal dendrites, and distal dendrites. Exogenously expressed Rab7TN, CA-LOV-Rab7TN-546-2, IE-LOV-Rab7TN-546-2, and PS-Rab7TN were also found in the soma, proximal dendrites, distal dendrites, and spines of PCs (Additional figure 6). We did not find clear differences in subcellular distribution between Rab7TN or IE-LOV-Rab7TN-546-2, an active form in terms of Rab7TN, and CA-LOV-Rab7TN-546-2, an inactive form in terms of Rab7TN. We therefore expect that there are no changes in subcellular distribution of PS-Rab7TN before and after photoactivation. Because there are no clear differences in subcellular distributions of different constructs, *we show only images of slices expressing Rab7TN or PS-Rab7TN, stained with an antibody for metabotropic glutamate receptor 1, which is concentrated on dendritic spines, in Supplemental figure 6 of revised manuscript.*

Reviewer #3 (Remarks to the Author):

Kim et al. report on the mechanisms underlying the maintenance of parallel fiber-Purkinje cell LTD. They propose and test a mechanism of LTD maintenance involving a reduction of the number of AMPA receptors available for trafficking into the postsynaptic membrane. They develop a photosensitive inhibitor of endosome sorting and also provide a simple model of AMPA receptor trafficking that supports and explains their findings. Their work suggests a mechanism for an LTD threshold and may have implications for the memory storage capacities of Purkinje cells. Overall the work is novel and clearly presented. The issue of how memories are stored in the cerebellum is an important one. I have just a few relatively minor comments.

Response:

We thank the reviewer for the encouraging comments regarding our manuscript as well as important suggestions for improving our manuscript. We revised our manuscripts, in accordance with the reviewer's suggestions.

1) I am unclear why Rab7TN, a blocker of vesicle sorting, completely blocks expression and maintenance of LTD (figure 3E). Isn't it expected that only the maintenance should be blocked? I thought expression was due to PKC and map kinase signaling. This should be discussed.

Response:

As the reviewer mentioned, we also expected that only the maintenance would be blocked by the expression of Rab7TN. We assume that long-term Rab7TN expression may have indirect effects and cause changes in other parts of the endosomal pathway, due to continuous interruption of sorting to LEs or lysosomes, as reported in the increased recycling of nutrient transporters (Edinger et al., 2003). Therefore, the expression of Rab7TN is not appropriate to precisely investigate when LE sorting is required for LTD. In contrast, PS-Rab7TN allows us to transiently inhibit LE sorting, and we found that the photoactivation at 0-5 min did not alter LTD, suggesting that a lack of transient depression in Purkinje cells expressing Rab7TN is due to an indirect effect of long-term Rab7TN expression.

In the revised manuscript, we included a discussion regarding possible indirect effects of Rab7TN expression (p.21, first 4 lines in 2nd paragraph).

2) A point relating to framing of results. At bottom of page 14 authors transition to more in depth analysis of the data involving the BLPA. The motivation for this might be more clear if the authors point out the fact that the blockade of LTD appears to be incomplete in all cases and they wanted to understand why. The more in depth analysis shows that this is because of two populations of responses, ie cells in which LTD is completely blocked or completely unaffected.

Response:

We appreciate this appropriate suggestion to clarify the flow. *We added sentences in the revised manuscript, as the reviewer suggested (p.16, first 4 lines of 2nd paragraph).*

3) Some brief discussion of functional implications might be useful, eg multi-state synapses have important computational advantages for memory storage capacity as

shown by Fusi and others. Mechanisms described here may relate to this.

Response:

We agree that it is an interesting functional implication. We found the variable timing of LE sorting, which appears to result from different conditions of synapses. Such different conditions of synapses would be associated with the “multiple states of synapses”. However, the cascade model built by Fusi et al. suggested the presence of transition between the states, while it is not yet known whether or not the synaptic conditions determining the timing of LE sorting would be altered by some stimulation. *We therefore briefly mentioned a possibility that variable conditions may imply multiple states of synapses, and cited a study by Fusi et al. (Fusi et al., 2005), in the Discussion of revised manuscript (p.25, lines 6-8).*

4) Under natural conditions LTD is likely induced by paired parallel fiber and climbing fiber stimulation. Did the authors try any of their manipulations on climbing fiber induced-LTD? If not, why and do they think the results would be the same?

Response:

We did not try to use the induction by paired stimulation of parallel fibers and a climbing fiber (PF&CF).

Different time course of LTD has been observed depending on the type of stimulus and stimulus intensity (Ito, 2001; Tanaka and Augustine, 2008). The kinetic model predicted that stimulus-dependent variability results from the efficacy of the stimulus in activating the PKC-MAPK positive feedback loop (Kuroda et al, 2001). In our previous study, we used PF& ΔV to induce LTD, and demonstrated that the positive feedback loop works for about 20 min to express LTD (Tanaka and Augustine, 2008). Because our motivation of the current study is to clarify what happens after this loop works for LTD expression, we used the same LTD induction protocol, PF& ΔV .

Because the precise time course of LTD expression varies according to the type of stimulus, stimulus frequency, or intensity, we expect that the timing when LE sorting works may be different if LTD is triggered by PF&CF. However, considering that there are no studies showing obvious differences in molecular mechanisms of LTD induction and expression between PF&CF and PF& ΔV triggered LTD, except for AMPAR activation at CF synapses, we think that both protocols of LTD stimulation require LE sorting for the transition from the expression to the maintenance phase.

In the revised manuscript, we added a reason why we used PF& ΔV in the Methods (p.28, lines 7-9), and added a possibility of different timing of LE sorting according to the type of stimuli in the Discussion (p.24, lines 16-17).

References for Responses to Reviewers' Comments

- Archibald, K., Perry, M.J., Molnar, E., and Henley, J.M. (1998). Surface expression and metabolic half-life of AMPA receptors in cultured rat cerebellar granule cells. *Neuropharmacology* 37, 1345-1353.
- Bucci, C., Thomsen, P., Nicoziani, P., McCarthy, J., and van Deurs, B. (2000). Rab7: a key to lysosome biogenesis. *Mol. Biol. Cell* 11, 467-480.
- Cohen, L.D., Zuchman, R., Sorokina, O., Muller, A., Dieterich, D.C., Armstrong, J.D., Ziv, T., and Ziv, N.E. (2013). Metabolic turnover of synaptic proteins: kinetics, interdependencies and implications for synaptic maintenance. *PLoS One* 8, e63191.
- Edinger, A.L., Cinalli, R.M., and Thompson, C.B. (2003). Rab7 prevents growth factor-independent survival by inhibiting cell-autonomous nutrient transporter expression. *Dev. Cell* 5, 571-582.
- Ehlers, M.D. (2000). Reinsertion or degradation of AMPA receptors determined by activity-dependent endocytic sorting. *Neuron* 28, 511-525.
- Erkens, M., Tanaka-Yamamoto, K., Cheron, G., Marquez-Ruiz, J., Prigogine, C., Schepens, J.T., Nadif Kasri, N., Augustine, G.J., and Hendriks, W.J. (2015). Protein tyrosine phosphatase receptor type R is required for Purkinje cell responsiveness in cerebellar long-term depression. *Mol. Brain* 8, 1.
- Feng, Y., Press, B., and Wandinger-Ness, A. (1995). Rab 7: an important regulator of late endocytic membrane traffic. *J. Cell Biol.* 131, 1435-1452.
- Fernandez-Monreal, M., Brown, T.C., Royo, M., and Esteban, J.A. (2012). The balance between receptor recycling and trafficking toward lysosomes determines synaptic strength during long-term depression. *J. Neurosci.* 32, 13200-13205.
- Fusi, S., Drew, P.J., and Abbott, L.F. (2005). Cascade models of synaptically stored memories. *Neuron* 45, 599-611.
- Jia, D., Zhang, J.S., Li, F., Wang, J., Deng, Z., White, M.A., Osborne, D.G., Phillips-Krawczak, C., Gomez, T.S., Li, H., *et al.* (2016). Structural and mechanistic insights into regulation of the retromer coat by TBC1d5. *Nat. Commun.* 7, 13305.
- Kjøller, C., and Diemer, N.H. (2000). GluR2 protein synthesis and metabolism in rat hippocampus following transient ischemia and ischemic tolerance induction. *Neurochem. Int.* 37, 7-15.
- Lee, D., Yamamoto, Y., Kim, E., and Tanaka-Yamamoto, K. (2015). Functional and Physical Interaction of Diacylglycerol Kinase ζ with Protein Kinase C α Is Required for Cerebellar Long-Term Depression. *J. Neurosci.* 35, 15453-15465.
- Matsuda, S., Kakegawa, W., Budisantoso, T., Nomura, T., Kohda, K., and Yuzaki, M. (2013). Stargazin regulates AMPA receptor trafficking through adaptor protein complexes during long-term depression. *Nat. Commun.* 4, 2759.
- Munsie, L.N., Milnerwood, A.J., Seibler, P., Beccano-Kelly, D.A., Tatarnikov, I., Khinda, J., Volta, M., Kadgien, C., Cao, L.P., Tapia, L., *et al.* (2015). Retromer-dependent neurotransmitter receptor trafficking to synapses is altered by the Parkinson's disease VPS35 mutation p.D620N. *Hum. Mol. Genet.* 24, 1691-1703.
- Romero Rosales, K., Peralta, E.R., Guenther, G.G., Wong, S.Y., and Edinger, A.L. (2009). Rab7 activation by growth factor withdrawal contributes to the induction of apoptosis. *Mol. Biol. Cell* 20, 2831-2840.
- Song, J.W., Misgeld, T., Kang, H., Knecht, S., Lu, J., Cao, Y., Cotman, S.L., Bishop, D.L., and Lichtman, J.W. (2008). Lysosomal activity associated with developmental axon pruning. *J. Neurosci.* 28, 8993-9001.
- Tanaka, K., and Augustine, G.J. (2008). A positive feedback signal transduction loop determines timing of cerebellar long-term depression. *Neuron* 59, 608-620.

- Tatsukawa, T., Chimura, T., Miyakawa, H., and Yamaguchi, K. (2006). Involvement of basal protein kinase C and extracellular signal-regulated kinase 1/2 activities in constitutive internalization of AMPA receptors in cerebellar Purkinje cells. *J. Neurosci.* 26, 4820-4825.
- Temkin, P., Morishita, W., Goswami, D., Arendt, K., Chen, L., and Malenka, R. (2017). The Retromer Supports AMPA Receptor Trafficking During LTP. *Neuron* 94, 74-82.
- Tian, Y., Tang, F.L., Sun, X., Wen, L., Mei, L., Tang, B.S., and Xiong, W.C. (2015). VPS35-deficiency results in an impaired AMPA receptor trafficking and decreased dendritic spine maturation. *Mol. Brain* 8, 70.
- Vonderheit, A., and Helenius, A. (2005). Rab7 associates with early endosomes to mediate sorting and transport of Semliki forest virus to late endosomes. *PLoS Biol.* 3, e233.
- Waung, M.W., Pfeiffer, B.E., Nosyreva, E.D., Ronesi, J.A., and Huber, K.M. (2008). Rapid translation of Arc/Arg3.1 selectively mediates mGluR-dependent LTD through persistent increases in AMPAR endocytosis rate. *Neuron* 59, 84-97.
- Yamamoto, Y., Lee, D., Kim, Y., Lee, B., Seo, C., Kawasaki, H., Kuroda, S., and Tanaka-Yamamoto, K. (2012). Raf kinase inhibitory protein is required for cerebellar long-term synaptic depression by mediating PKC-dependent MAPK activation. *J. Neurosci.* 32, 14254-14264.

Additional figures

Additional figure 1: Diffusion of intracellular solutions measured by Alexa dye imaging

Alexa568 included in the internal solution together with TeTx was imaged at the Purkinje cell dendrites where PFs are stimulated, while PF-EPSCs were recorded. The time course of increase in the Alexa568 intensity (red circles) was plotted together with normalized PF EPSC amplitudes (filled circles). The time to reach half-maximum in Alexa568 intensity was 15 min, extracted by fitting an exponential growth function (blue line).

Additional figure 2: The model including the assumption of slow kinetics (the half decay period of 15 min) of exocytosis blockade

A, The model was modified by considering an assumption that exocytosis blockade linearly depends on the concentration of TeTx, which is diffused in Purkinje cells with a time constant of 15 min. Thus, the decrease in k_{exo} follows exponential decay with half decay period of 15 min (blue line). B, In order to reproduce the experimental results of TeTx-mediated PF-EPSC reduction, an extremely large k_{endo} value (200 min^{-1}) (blue solid line) needed to be used.

Additional figure 3: Estimation of basal AMPAR endocytosis rate by the model with the assumption of slow kinetics (the half decay period of 15 min) of exocytosis blockade

In the model with slow kinetics of exocytosis blockade, shown as the blue solid line in additional figure 2B, the sudden reduction of k_{exo} (black line) allows us to estimate the basal AMPAR endocytosis rate in this trafficking system. The estimated rate was $T_{1/2} = 0.7$ min.

Additional figure 4: A limited impact of addition of secretory pathway on the k_{exo} inhibition-mediated EPSC reduction in the model

A, The secretory pathway ($k_{exo-sec} = 3 \times 10^{-4} \text{ min}^{-1}$) was included in the model. B, Time course of EPSC reduction by the inhibition of exocytosis in the model with secretory pathway. The solid lines show the results obtained by the inhibition of both k_{exo} and $k_{exo-sec}$ (color lines, $k_{exo-sec} = 3 \times 10^{-5} \text{ min}^{-1}$) or by inhibition of k_{exo} alone (black and gray lines). For comparison, the results of original model without secretory pathway are also shown (dotted lines). Note that three lines are mostly overlapped. C, Changes in EPSC level (top), and changes in $T_{1/2}$ (middle) or $MR_{PF-EPSC}$ (bottom) of EPSC reduction by the inhibition of both k_{exo} and $k_{exo-sec}$, when N_{tot} is reduced (left) or k_{endo} is increased (right) in the model with secretory pathway.

Additional figure 5: Inability of Rab7TN to bind with RBD-RILP

Immunoblots of GFP-Rab7, GFP-Rab7TN, and GFP-Rab7QL (constitutive active form of Rab7), which were bound with GST-fused RBD-RILP in the GST pull-down assay (PD), or were expressed in the HEK293 cell lysates used for the pull-down assay (input). While Rab7WT and Rab7QL bound with RBD-RILP, Rab7TN did not bind with RBD-RILP.

Additional figure 6: AAV-mediated expression of Rab7TN constructs in Purkinje cells

Confocal images of cerebellar slices obtained from mice subjected to the stereotaxic injection of AAV expressing GFP-fused (green) Rab7TN (A), PS-Rab7TN (B), IE-(C) or CA-LOV-Rab7TN-546-2 (D). Slices were stained with an antibody of mGluR1 (red), which is concentrated on dendritic spines of Purkinje cells. Scale bars, 40 μm (top) or 10 μm (bottom).

Responses to issues raised by referees

We truly appreciate reviewers for positive comments on our manuscript and for their efforts to further improve it. Here are our responses to the remaining suggestions provided by reviewer #1.

Reviewer #1 (Remarks to the Author):

1. I agree with the explanations provided concerning the diffusion of TeNT into cells and the comparison of rates of decrease to determine k_{endo} and N_{tot} . However, I think adding the data in additional figure 1 of the rebuttal letter to a supplementary figure in the manuscript would help the reader to know exactly how the measurements were obtained.

We thank the reviewer for agreeing our explanations.

According to the suggestion provided by the reviewer, we included the figure, which shows the time course of PF-EPSC reduction by TeTx and of diffusion of internal solutions (previous additional figure 1), in Supplemental Fig. 1a, and described this figure in the Results as well as the Methods (p. 5, 5th line from the bottom, p. 7, lines 9-10 & p. 25, line 15).

4. The modelling and analysis is now much clearer. Supplementary Figure 9 helps to understand the interplay between the increase in endocytosis and the transition from early to late endosomes. Perhaps in the discussion the authors could refer to two recent publications showing that an LTD protocol in hippocampal cultured neurons transiently increases AMPAR endocytosis, similar to the prediction of the model (Rosendale et al. Cell Reports 2017; Fujii et al. Genes Cells 2017).

We agree that results of these papers, which demonstrated transient increase in AMPAR endocytosis by LTD stimulation in hippocampal cultured neurons, are in line with the model based on our experimental results. Due to the word limitation, we simply referred to these papers at the end of discussion, in which we mentioned about similarity of hippocampal LTD with cerebellar LTD (p. 22, 4th line from the bottom).